# Stromal PTEN determines mammary epithelial response to radiotherapy

Gina M. Sizemore[1,2], Subhasree Balakrishnan [1,3], Katie A. Thies[4,5], Anisha M. Hammer[1,6], Steven T. Sizemore [1,2], Anthony J. Trimboli[4,5], Maria C. Cuitiño[4,5], Sarah A. Steck[1], Gary Tozbikian[7], Raleigh D. Kladney[1], Neelam Shinde[1], Manjusri Das[1], Dongju Park[1,3], Sarmila Majumder[1], Shiva Krishnan[1,8], Lianbo Yu[9], Soledad A. Fernandez[9], Arnab Chakravarti[1,2], Peter G. Shields[1,8], Julia R. White[1,2], Lisa D. Yee[10], Thomas J. Rosol[11], Thomas Ludwig[1,3], Morag Park[12], Gustavo Leone[4,5] & Michael C. Ostrowski [4,5]

The importance of the tumor–associated stroma in cancer progression is clear. However, it remains uncertain whether early events in the stroma are capable of initiating breast tumorigenesis. Here, we show that in the mammary glands of non-tumor bearing mice, stromal-specific phosphatase and tensin homolog (*Pten*) deletion invokes radiation-induced genomic instability in neighboring epithelium. In these animals, a single dose of whole-body radiation causes focal mammary lobuloalveolar hyperplasia through paracrine epidermal growth factor receptor (EGFR) activation, and EGFR inhibition abrogates these cellular changes. By analyzing human tissue, we discover that stromal PTEN is lost in a subset of normal breast samples obtained from reduction mammoplasty, and is predictive of recurrence in breast cancer patients. Combined, these data indicate that diagnostic or therapeutic chest radiation may predispose patients with decreased stromal PTEN expression to secondary breast cancer, and that prophylactic EGFR inhibition may reduce this risk.

[1] The Comprehensive Cancer Center, The Ohio State University, Columbus, OH 43210, USA. [2] Department of Radiation Oncology, The Ohio State University, Columbus, OH 43210, USA. [3] Department of Cancer Biology and Genetics, The Ohio State University, Columbus, OH 43210, USA. [4] Hollings Cancer Center, Medical University of South Carolina, Charleston, SC 29425, USA. [5] Department of Biochemistry & Molecular Biology, Medical University of South Carolina, Charleston, SC 29425, USA. [6] Division of Endocrinology, Diabetes and Metabolism, Department of Internal Medicine, The Ohio State University, Columbus 43210 OH, USA. [7] Department of Pathology, The Ohio State University Wexner Medical Center, Columbus 43210 OH, USA. [8] Department of Internal Medicine, College of Medicine, The Ohio State University, Columbus, OH 43210, USA. [9] Department of Biomedical Informatics' Center for Biostatistics, The Ohio State University, Columbus, OH 43210, USA. [10] Division of Surgical Oncology, Department of Surgery, City of Hope, Duarte, CA 91010, USA. [11] Department of Molecular and Cellular Biology, College of Arts and Sciences, Ohio University, Athens, OH 45701, USA. [12] Rosalind and Morris Goodman Cancer Research Centre, McGill University, Montréal H3A 1A3 QC, Canada. Correspondence and requests for materials should be addressed to G.L. (email: leoneg@musc.edu) or to M.C.O. (email: ostrowsk@musc.edu)

Stroma in the breast tumor microenvironment (TME) is comprised of perivasculature, endothelia, fibroblasts, adipocytes, and immune cells[1,2]. These cells can be reprogrammed to promote tumor growth as well as metastatic spread. Previous work by our laboratory described the tumor suppressive function of phosphatase and tensin homolog (PTEN) signaling in the breast TME[3]. Acting as a phosphatidylinositol-3,4,5 trisphosphate 3-phosphatase, PTEN is a well-described tumor suppressor that deactivates phosphoinoside substrates required for signaling by phosphatidylinositol 3-kinase (PI3K), a key component of the AKT/PKB survival pathway[4]. To directly study stromal-specific PTEN function, we developed and utilized a transgene encoding *Cre* recombinase downstream of the fibroblast specific-1 promoter (*Fsp1/S100A4*) to conditionally delete PTEN in cells of mesenchymal origin[3]. PTEN was deleted in the mammary stroma of a HER2 (human epidermal growth factor receptor 2) mouse model of breast cancer (MMTV-*ErbB2*)[5] revealing accelerated appearance and progression of *ErbB2*-induced tumors with invasive properties[3]. In this model, loss of stromal PTEN hyperactivates the transcription factor ETS2 (v-Ets avian erythroblastosis virus E26 oncogene homologue 2), ultimately generating an oncogenic secretome, and increasing extracellular matrix (ECM) remodeling, immune cell infiltration, and tumor vascularization[3,6]. While these findings have brought recognition that an active stroma participates in all phases of cancer progression[1,2,7–10], whether a pro-oncogenic TME co-evolves in response to cues from neighboring transformed breast epithelia, or whether changes within the normal microenvironment can be sufficient to initiate malignant transformation of the breast is still unclear.

In the current study, we evaluated how genetic loss of stromal PTEN alters the mammary epithelium prior to tumor formation and discovered an unexpected role for stromal PTEN function in epithelial genetic stability. Comparison of gene expression in non-tumor bearing control *ErbB2;Pten^{fl/fl}* and experimental *ErbB2;Fsp-cre;Pten^{fl/fl}* mice uncovered a defect in DNA repair in pre-neoplastic ductal luminal epithelium of mammary glands lacking stromal PTEN. Mechanistically, PTEN-null stroma hyperactivates epidermal growth factor receptor (EGFR)/ErbB2 signaling in neighboring epithelial cells. This epithelium is sensitive to radiation-induced DNA damage and more readily accumulates centrosome amplification and chromosomal aberrations. Importantly, pharmacological inhibition of EGFR is sufficient to prevent chromosomal instability. Consistent with these experimental findings, there is wide inter- and intra-individual variation in breast stromal PTEN levels in healthy women, raising the possibility that low breast stromal PTEN alters a patient's response to radiation due to paracrine EGFR-induced epithelial genomic instability. Further examination of HER2-positive breast cancer patients treated with radiation showed that stromal PTEN status in adjacent normal tissue predicts recurrence when considered with estrogen receptor-α (ER) status. These findings indicate that normal breast tissue with low stromal PTEN is at risk for transformation when exposed to DNA-damaging agents. Indeed, a single dose of DNA-damaging whole-body radiation is sufficient to induce mammary hyperplasia in mice with PTEN-null stroma. Blocking EGFR prior to radiation inhibited these cellular changes in our mouse model suggesting prophylactic use of EGFR inhibitors could reduce secondary radiation-induced malignancies in women receiving chest radiation.

## Results

### Stromal PTEN maintains epithelial DNA repair response.
To determine whether stromal PTEN deletion exerts stable pro-tumorigenic effects on neighboring mammary epithelium, control (*ErbB2;Pten^{fl/fl}*) and experimental (*ErbB2;Fsp-cre;Pten^{fl/fl}*) mice were sacrificed prior to exhibiting observable neoplasia (~8–10 weeks of age) and their mammary epithelium was injected at a limiting dose ($1 \times 10^5$ cells) into the mammary fat pads of wild-type syngeneic recipients and monitored for tumor development. Mammary epithelium was purified by CD24/CD29 dual positivity (upper right quadrant)[11,12] to exclude PTEN-null fibroblast contamination from being transferred to syngeneic recipient mice (Fig. 1a; Supplementary Fig. 1a). There were no overt differences in the CD24+/CD29+ (upper right quadrant) bulk epithelial populations between control and experimental tissue (Fig. 1a versus Supplementary Fig. 1a), which is important to note given that we have previously reported an increase in the CD24+/CD29^{hi} mammary stem cell enriched (MaSC-enriched) subpopulation in mice with stromal PTEN deletion[12]. Over the course of one year, mammary tumors only arose in recipient mice receiving *ErbB2;Fsp-cre;Pten^{fl/fl}* donor epithelium, suggesting a permanent, pro-tumorigenic effect elicited by neighboring PTEN-null stroma prior to injection (Fig. 1b, c). These tumors maintain PTEN expression as indicated by the lack of a *Pten* deleted allele (Supplementary Fig. 1b), the maintenance of *Pten* mRNA (Supplementary Fig. 1c-top), the lack of detectable *Cre* mRNA (Supplementary Fig. 1c-bottom), the maintenance of PTEN immunostaining (Supplementary Fig. 1d-left) and the lack of X-gal positivity as a readout of *Cre* expression (Supplementary Fig. 1d-right). Combined, these supporting data confirm *Fsp-cre* is not being aberrantly activated in this tumor tissue through epithelial-to-mesenchymal transition (EMT) or other means resulting in *Pten* deletion post-transplant. These data are consistent with our previous work indicating a lack of EMT in *ErbB2; Fsp-cre;Rosa^{loxP}* tumors[13].

To begin to understand how stromal PTEN exerts this oncogenic effect, pre-neoplastic mammary tissue was purified into well-characterized epithelial subpopulations (MaSC-enriched, luminal progenitor, and mature luminal)[11,12] for gene expression analysis (Fig. 1d). Maintenance of PTEN mRNA and protein expression in the sorted *ErbB2;Fsp-cre;Pten^{fl/fl}* mature luminal epithelial population was confirmed by qRT-PCR and immunofluorescence, respectively (Supplementary Fig. 2a, b). Unsupervised gene set enrichment analysis (GSEA) of the curated C5 gene sets within the Molecular Signatures Database (MSigDB) revealed significant (false discovery rate (FDR) $q < 0.05$) de-enrichment of gene sets involved in DNA repair and mitotic cell cycle control in the *ErbB2;Fsp-cre;Pten^{fl/fl}* mature luminal population (Supplementary Table 1; Fig. 1e; Supplementary Fig. 2c), which are the cells that ultimately generate MMTV-*ErbB2* luminal tumors[5,14–16]. The observed decrease in DNA repair genes corresponds to de-enrichment of base excision repair, homologous recombination, mismatch repair, and non-homologous end joining repair processes (Supplementary Fig. 2d), while the decrease in cell cycle related gene expression is specific to M-phase (Supplementary Fig. 2e). To determine the functional significance of these gene expression changes, epithelial cells from control (*ErbB2;Pten^{fl/fl}*) and experimental (*ErbB2;Fsp-cre;Pten^{fl/fl}*) mice were isolated by gravity separation and subjected to X-ray radiation in vitro to induce DNA damage. Cells were harvested at 6 h post-radiation and evaluated for RAD51 (Fig. 1f) and γ-H2AX foci (Fig. 1g). Non-irradiated control cells were harvested in parallel to evaluate baseline levels of DNA damage (Supplementary Fig. 3a). Control epithelium showed substantial DNA damage-induced RAD51 foci, and thus, not surprisingly, limited γ-H2AX foci at 6 h post-radiation indicating efficient DNA repair. In contrast, epithelium from *ErbB2;Fsp-cre;Pten^{fl/fl}* mice exhibit decreased RAD51 and persistent γ-H2AX foci indicating a failure in the DNA damage response (Fig. 1f, g).

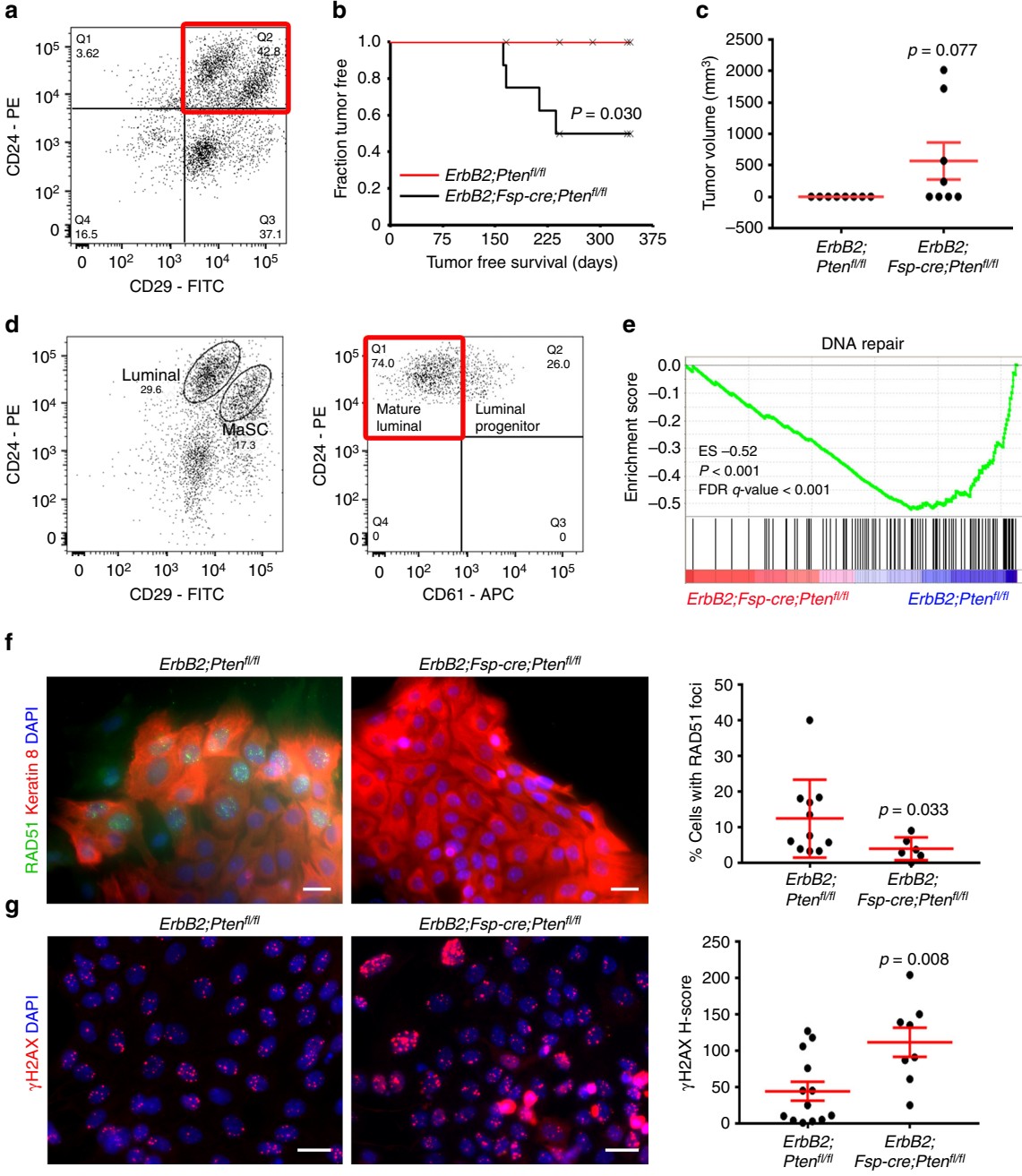

**Fig. 1** Loss of stromal PTEN decreases the DNA repair response in associated mammary epithelium. **a** Representative FACS plot defining CD24+CD29+ mammary epithelium (*ErbB2;Fsp-cre;Pten^{fl/fl}*). **b** Tumor-free survival of wild-type FVB/N mice injected orthotopically with *ErbB2;Fsp-cre;Pten^{fl/fl}* versus control *ErbB2;Pten^{fl/fl}* epithelium (*n* = 8/genotype). *P* value determined by Log-rank (Cox-Mantel). **c** Tumor volume (mean ± s.e.m.) at time of harvest after orthotopic injection of *ErbB2;Fsp-cre;Pten^{fl/fl}* (*n* = 8) versus control *ErbB2;Pten^{fl/fl}* (*n* = 7) epithelium. *P* value determined by Fisher's exact. **d** Representative FACS plot defining *ErbB2;Pten^{fl/fl}* mammary epithelial subpopulations segregated by CD29 and CD24 (left: luminal and mammary stem cell (MaSC)) and CD61 (right: mature luminal and luminal progenitor). **e** Gene set enrichment analysis (GSEA) for DNA repair genes in *ErbB2;Fsp-cre;Pten^{fl/fl}* versus control *ErbB2;Pten^{fl/fl}* mature luminal epithelium. **f** Representative RAD51/keratin 8 dual immunofluorescence and quantification (mean ± s.e.m.) of epithelial cells isolated from *ErbB2;Pten^{fl/fl}* versus *ErbB2;Fsp-cre;Pten^{fl/fl}* mice irradiated (3 Gy) in vitro and evaluated 6 h post-radiation. *P* value determined by Welch's *t*-test (*ErbB2;Pten^{fl/fl}* *n* = 11 fields/3 mice, *ErbB2;Fsp-cre;Pten^{fl/fl}* *n* = 6 fields/2 mice). Scale bar = 20 μm. **g** Representative γ-H2AX immunofluorescence and quantification (mean ± s.e.m.) of epithelial cells isolated from *ErbB2;Pten^{fl/fl}* and *ErbB2;Fsp-cre;Pten^{fl/fl}* mice irradiated (3 Gy) in vitro and evaluated 6 h post-radiation. *P* value determined by an unpaired, two-tailed Student's *t* test (*ErbB2;Pten^{fl/fl}* *n* = 13 fields/3 mice, *ErbB2;Fsp-cre;Pten^{fl/fl}* *n* = 8 fields/3 mice). Scale bar = 20 μm

To test whether stromal PTEN deletion induces a similar defect in the intact mammary gland, mice were exposed to a single dose (6 Gray (Gy)) of whole-body X-ray radiation and evaluated for epithelial γ-H2AX. At 30 min post-radiation, γ-H2AX recruitment to damaged DNA was pronounced irrespective of genotype,

confirming a radiation-induced response in this tissue (Supplementary Fig. 3b). At 6 h post-radiation, control (*Pten^{fl/fl}* and *ErbB2;Pten^{fl/fl}*) mammary glands resolved most of their DNA damage as indicated by minimal γ-H2AX staining (Fig. 2a). In contrast, both *Fsp-cre;Pten^{fl/fl}* and *ErbB2;Fsp-cre;Pten^{fl/fl}* glands

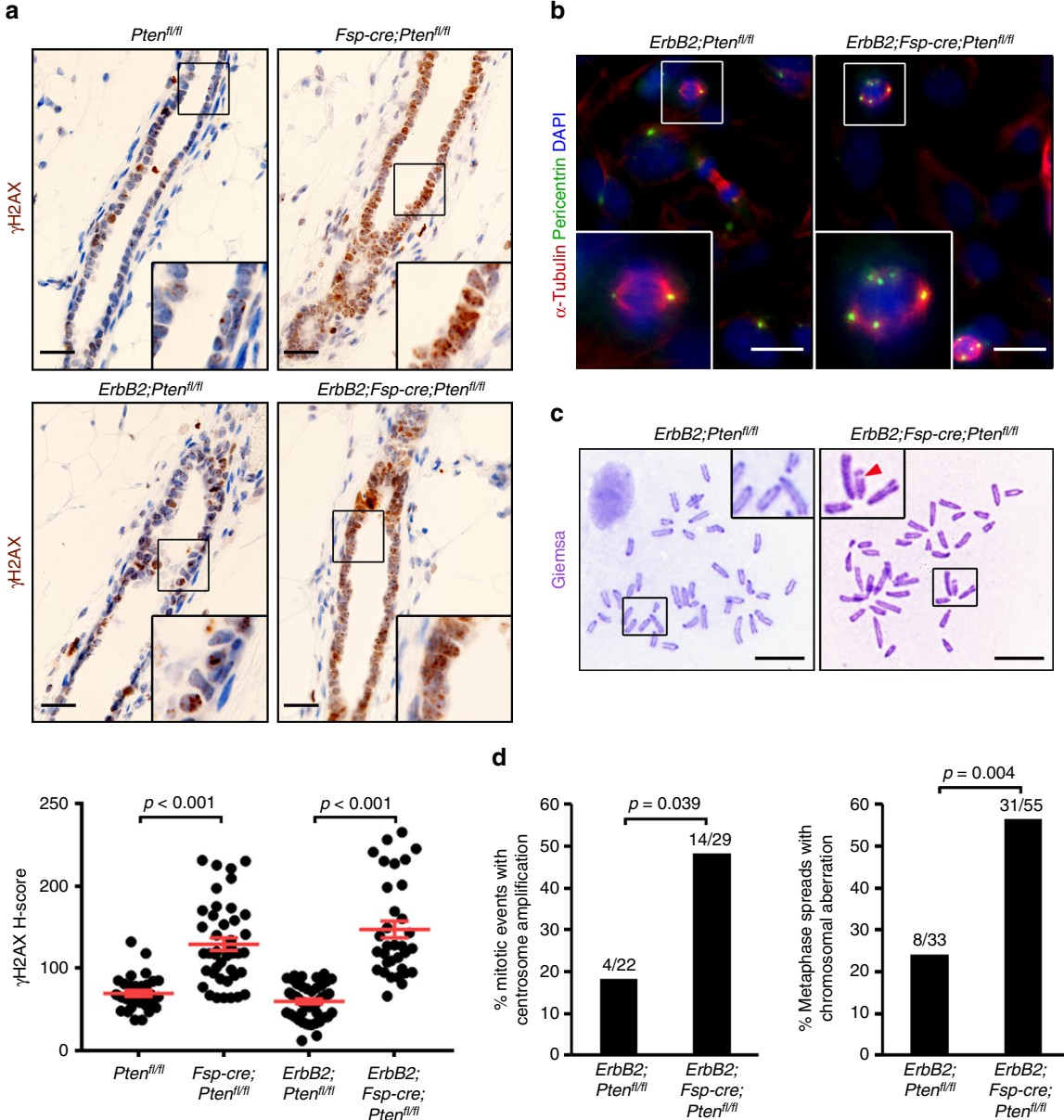

**Fig. 2** Loss of stromal PTEN increases genomic instability in associated epithelium. **a** Representative γ-H2AX immunohistochemistry and quantification (mean ± s.e.m.) of mammary epithelium in $Pten^{fl/fl}$, $Fsp$-$cre;Pten^{fl/fl}$, $ErbB2;Pten^{fl/fl}$ and $ErbB2;Fsp$-$cre;Pten^{fl/fl}$ mice irradiated (6 Gy whole-body) and evaluated 6 h post-radiation. Both $p$ values determined by unpaired, two-tailed Student's $t$ test ($Pten^{fl/fl}$ $n = 32$ fields/3 mice, $Fsp$-$cre;Pten^{fl/fl}$ $n = 42$ fields/3 mice, $ErbB2;Pten^{fl/fl}$ $n = 47$ fields/5 mice, $ErbB2;Fsp$-$cre;Pten^{fl/fl}$ $n = 33$ fields/3 mice). Scale bar = 20 μm. **b** Representative α-Tubulin/pericentrin dual immunofluorescence of epithelial cells isolated from $ErbB2;Pten^{fl/fl}$ and $ErbB2;Fsp$-$cre;Pten^{fl/fl}$ mice irradiated (6 Gy whole-body) and evaluated 1 week post-radiation in vitro. Scale bar = 25 μm. **c** Representative metaphase spreads of epithelial cells isolated from $ErbB2;Pten^{fl/fl}$ and $ErbB2;Fsp$-$cre;Pten^{fl/fl}$ mice irradiated (6 Gy whole-body) and evaluated 1 week post-radiation in vitro. Arrowhead = chromosome break. Scale bar = 10 μm. **d** Quantification of centrosome amplification (shown in **b**) and chromosomal aberrations (shown in **c**) in epithelial cells isolated from $ErbB2;Pten^{fl/fl}$ and $ErbB2;Fsp$-$cre;Pten^{fl/fl}$ mice irradiated (6 Gy whole-body) and evaluated 1 week post-radiation in vitro. Both $p$ values determined by two-tailed, Fisher's exact test (left: $ErbB2;Pten^{fl/fl}$ $n = 22$ fields/2 mice, $ErbB2;Fsp$-$cre;Pten^{fl/fl}$ $n = 29$ fields/2 mice; right: $ErbB2;Pten^{fl/fl}$ $n = 33$ spreads/4 mice, $ErbB2;Fsp$-$cre;Pten^{fl/fl}$ $n = 55$ spreads/4 mice)

continued to exhibit high levels of epithelial γ-H2AX, indicating that mammary epithelial DNA repair is dependent upon PTEN status in the associated stroma and is not effected by expression of the $ErbB2$ oncogene (Fig. 2a). Notably, while radiation-induced overt stromal DNA damage is dramatically lower than in the associated epithelium regardless of genotype (i.e., the $Pten^{fl/fl}$ stromal H-score mean is 4.5 (Supplementary Fig. 3c) versus an epithelial H-score mean of 69.6 (Fig. 2a) in the same tissue), stromal-specific γ-H2AX staining is significantly ($p \leq 0.001$ by

two-tailed Mann–Whitney) increased with stromal PTEN loss (Supplementary Fig. 3c). These results confirm the known cell autonomous role for PTEN in genomic stability[17–20]. Importantly, the observed DNA damage phenotype is not a result of $Cre$ expression alone[21,22] (Supplementary Fig. 3d) and occurred independent of changes in proliferation and apoptosis (Supplementary Fig. 3e, f).

Since GSEA also predicted de-enrichment in mitotic cell cycle related gene expression in mature luminal epithelium derived

from mammary glands with PTEN-null stroma (Supplementary Figs. 2c, e), we further evaluated whether the observed DNA repair defects were sustained over time. To this end, control (*ErbB2; Pten^fl/fl*) and experimental (*ErbB2;Fsp-cre;Pten^fl/fl*) mice were similarly irradiated and allowed to recover for 1 week, at which time the epithelium was harvested, evaluated for pericentrin and α-tubulin localization to quantify centrosome amplification (Fig. 2b), and karyotyped to evaluate chromosomal aberrations (Fig. 2c). Centrosome amplification was almost tripled and the number of chromosomal aberrations were more than doubled in epithelial cells derived from *ErbB2;Fsp-cre;Pten^fl/fl* compared to cells from control (*ErbB2;Pten^fl/fl*) mice (Fig. 2d), suggesting that compromised DNA repair in mammary glands lacking stromal PTEN leads to quantifiable genomic instability.

**Stromal PTEN maintains epithelial genomic instability**. Stromal PTEN deletion presumably necessitates a non-cell autonomous mechanism to induce alterations within the epithelial compartment. It is known that the epithelium in stromal PTEN-null mammary glands exhibits enhanced AKT, JNK, and ERK signaling,[3] and that the ErbB family of receptor tyrosine kinases signals through these effectors[23]. Given that both amplified HER2[24–26] and hyperactivated EGFR[27] induce genomic instability, we hypothesized that loss of stromal PTEN could hyperactivate ErbB signaling in adjacent epithelial cells through increased ErbB ligand production, which would subsequently lead to the observed genomic instability. To test this hypothesis directly, immortalized mouse mammary fibroblasts (MMFs) isolated from either control or stromal PTEN-null mice were evaluated for mRNA expression of ErbB ligands. While the majority of the neuregulins were unchanged or undetectable, the EGF ligands amphiregulin (*Areg*), betacellulin (*Btc*), epiregulin (*Ereg*), hb-EGF (*Hbegf*), and TGF-α (*Tgfa*) were all increased in mammary fibroblasts lacking PTEN (Fig. 3a). Upregulation of TGF-α, amphiregulin and epiregulin protein were confirmed by immunoblotting (Fig. 3b) or enzyme-linked immunosorbent assays (ELISAs) (Supplementary Fig. 4a). ErbB2 is ligandless. As such, this selective increase in EGF ligands strongly indicates EGFR as the primary ErbB family member responsive to the PTEN-null stromal milieu.

To determine if the observed increase in EGF ligands was functionally relevant, mammary glands from control (*Pten^fl/fl*) and experimental (*Fsp-cre;Pten^fl/fl*) mice were evaluated for increased activation of both EGFR and ErbB2, given that EGFR/EGFR homo-dimers and EGFR/ErbB2 hetero-dimers would be anticipated upon EGFR activation[28,29]. Both epithelial phospho-ErbB2 (Tyr1221/22) ($p = 0.001$ by two-tailed Mann–Whitney) and phospho-EGFR (Tyr1068) ($p = 0.046$ by Welch's *t*-test) were significantly increased in mammary glands lacking stromal PTEN (Fig. 3c, d). Specificity and localization of the phospho-EGFR (Tyr1068) antibody was further confirmed by immunofluorescent staining (Supplementary Fig. 4b). These findings suggest that loss of stromal PTEN is associated with increased EGF ligand production and EGFR/ErbB2 hyperactivation in adjacent epithelial cells.

To test whether increased EGFR/ErbB2 signaling is causal for the observed DNA repair defect, control (*Pten^fl/fl*) and experimental (*Fsp-cre;Pten^fl/fl*) mice were pre-treated with vehicle (DMSO) or the selective EGFR small molecule inhibitor erlotinib for 1 week. Mice were then irradiated (6 Gy whole-body) and mammary tissue was harvested 6 h post-radiation. Remarkably, pre-treatment with erlotinib significantly ($p < 0.001$ by two-tailed Mann–Whitney) reduced the amount of epithelial γ-H2AX staining in the *Fsp-cre;Pten^fl/fl* mammary glands when compared to glands of vehicle treated mice (Fig. 4a). Diminished phospho-

EGFR expression was confirmed in control (*Pten^fl/fl*) epithelium as a response to erlotinib treatment (Fig. 4b), and importantly, EGF ligand mRNA expression is not reduced in isolated PTEN-null (*Fsp-cre;Pten^fl/fl*) MMFs with in vitro EGFR inhibition supporting a direct epithelial-specific response (Supplementary Fig. 5a).

Erlotinib selectivity for EGFR is known[30–32]; however as mentioned above, EGFR/ErbB2 hetero-dimerization[29] is likely in mammary epithelium prompting us to also evaluate ErbB2 activity after erlotinib treatment. In contrast to a reduction in phospho-EGFR (Fig. 4b), no detectable change in phospho-ErbB2 expression was observed in the control (*Pten^fl/fl*) epithelium (Supplementary Fig. 5b). As a way to directly test ErbB2 function in the observed DNA damage response, we also performed a paralleling experiment using a selective ErbB2 inhibitor (CP-724,714)[33]. Interestingly, similar to erlotinib, the CP-724,714 pre-treatment was effective at significantly ($p < 0.001$ by two-tailed Student's *t*-test) rescuing the DNA damage defect observed in mice with stromal PTEN deletion (Supplementary Fig. 6a). As expected, a significant ($p < 0.009$ by two-tailed Mann–Whitney) reduction in phospho-ErbB2 levels by CP-724,714 was observed at 6 h post-radiation in control (*ErbB2;Pten^fl/fl*) epithelium (Supplementary Fig. 6b). Importantly, EGFR activity was significantly ($p < 0.001$ by two-tailed Student's *t*-test) reduced by CP-724,714 at 30 min post-radiation in control (*ErbB2;Pten^fl/fl*) epithelium (Supplementary Fig. 6c) providing further support of EGFR's involvement in the observed DNA repair defect. To further confirm this hypothesis, we then evaluated whether pre-treatment with erlotinib could negate the observed genomic instability in these mammary glands in response to radiation (Fig. 2b–d). Indeed, erlotinib pre-treatment of *ErbB2;Fsp-cre; Pten^fl/fl* female mice prevented the radiation-induced accumulation of chromosomal aberrations (Fig. 4c). Thus, inhibition of EGFR signaling in an experimental mouse model predisposed to genomic instability by loss of stromal PTEN returned the number of chromosomal lesions to baseline levels found in control animals.

**Stromal PTEN loss is observed in normal breast tissue**. To determine the clinical relevance of stromal PTEN in maintaining mammary epithelial genomic stability, we evaluated PTEN expression in the breast tissues of women who underwent reduction mammoplasty[34–36]. Notably, there was wide inter-individual variation in the range of stromal PTEN levels measured by immunohistochemistry, including some women who had ducts essentially devoid of any detectable stromal PTEN protein (Fig. 5a). There was also a wide range of intra-sample heterogeneity (Supplementary Fig. 7a), where a subset of samples had regions with both high stromal PTEN (maximum H-score field per sample indicated by the gray line) and low stromal PTEN (minimum H-score field per sample indicated by the red line). Given our findings above, we hypothesized that focal areas lacking stromal PTEN within the breast may be primed for tumor initiation in response to a DNA-damaging insult, such as radiation. Radiation does in fact increase the risk for secondary cancers, including breast cancer[37,38]. To begin to test this postulate, we evaluated stromal PTEN in tumor-adjacent normal epithelium in HER2-positive breast cancer patients treated with radiation ($n = 43$) (Fig. 5b; Supplementary Table 2). While overall stromal PTEN levels were not associated with breast cancer recurrence (Fig. 5c), there was a strong association of low stromal PTEN expression with recurrence (combined local and distant) in women having HER2-positive/ER-positive tumors (Fig. 5d). This is important because ER positivity is associated with a more favorable outcome in HER2 disease[39–42] (Supplementary Fig. 7b).

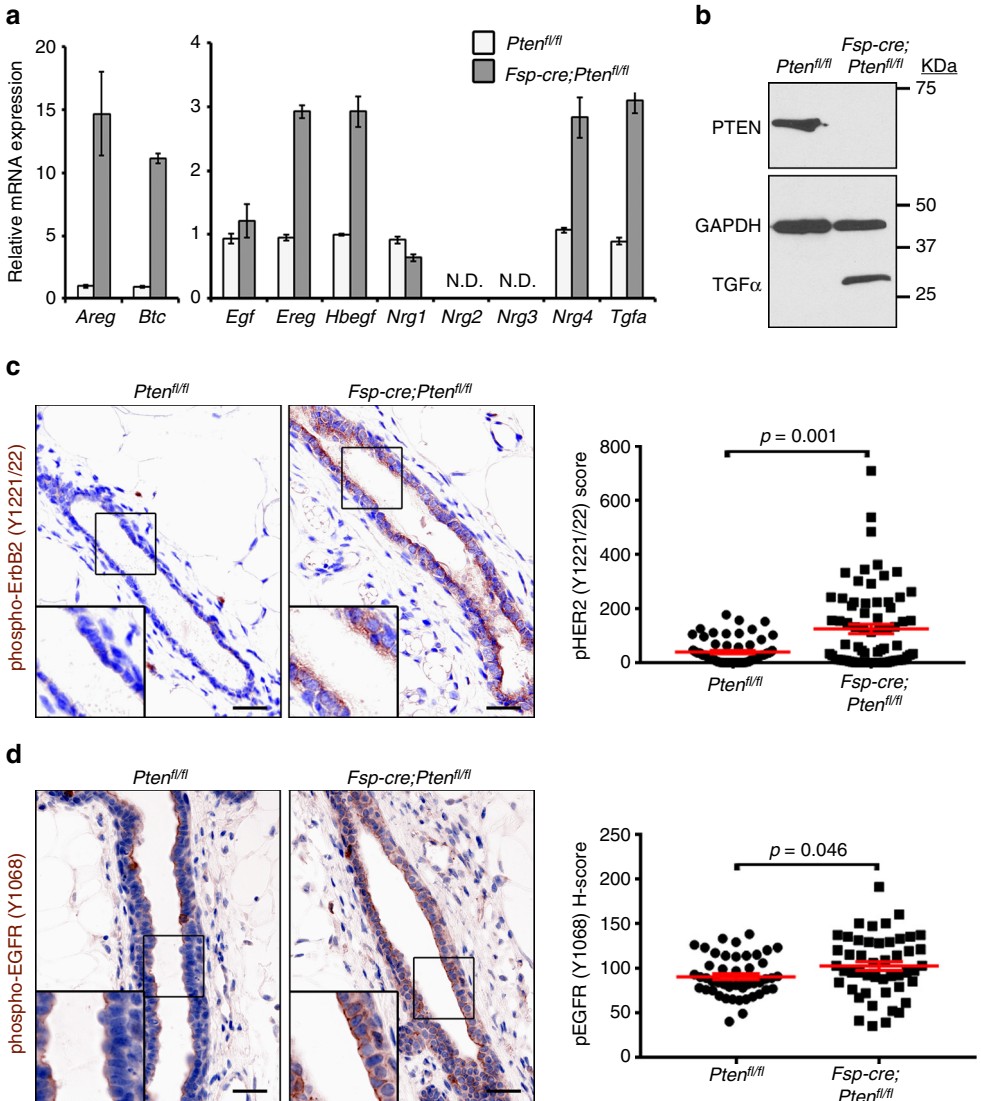

**Fig. 3** Loss of stromal PTEN increases fibroblast EGF ligand expression and associated ErbB signaling activity in neighboring epithelium. **a** *Areg*, *Btc*, *Egf*, *Ereg*, *Hbegf*, *Nrg1*, *Nrg4* and *Tgfa* mRNA in immortalized control (*Pten^{fl/fl}*) and PTEN-null (*Fsp-cre;Pten^{fl/fl}*) MMFs. Bars represent mean expression of technical replicates relative to *Gapdh* ± s.e.m. *Nrg2* and *Nrg3* were not detectable (N.D.). **b** Western blot for PTEN, TGF-α and GAPDH in the immortalized control (*Pten^{fl/fl}*) and PTEN-null (*Fsp-cre;Pten^{fl/fl}*) MMFs. **c** Representative phospho-ErbB2 (Y1221/22) immunostaining and quantification (mean ± s.e.m.) of mammary epithelium in *Pten^{fl/fl}* and *Fsp-cre;Pten^{fl/fl}* mice. *P* value determined by two-tailed Mann–Whitney (*Pten^{fl/fl}* n = 61 fields/4 mice, *Fsp-cre;Pten^{fl/fl}* n = 71 fields/4 mice). Scale bar = 20 μm. **d** Representative phospho-EGFR (Y1068) immunostaining and quantification (mean ± s.e.m.) of mammary epithelium in *Pten^{fl/fl}* and *Fsp-cre;Pten^{fl/fl}* mice. *P* value determined by Welch's *t*-test (*Pten^{fl/fl}* n = 48 fields/3 mice, *Fsp-cre;Pten^{fl/fl}* n = 48 fields/3 mice). Scale bar = 20 μm

To test whether reduced stromal PTEN expression may alter EGF ligand expression in cancer associated fibroblasts (CAFs) derived from breast cancer patients, we isolated breast CAFs from a patient at the time of surgery and used lentiviral knockdown to deplete PTEN. In this sample, loss of PTEN increased *EREG* and *HBEGF* expression (Fig. 5e). Induction of EGF ligands (*BTC*, *EGF*, and *EREG*) was similarly observed in fibroblasts from adjacent normal tissue (>10 cm from tumor) in a second patient (Fig. 5f). As seen in mouse MMFs (Fig. 3a), loss of PTEN did not consistently alter neuregulin gene expression (Supplementary Fig. 7c, d). Combined, these data indicate that stromal PTEN impacts EGFR signaling in both mouse mammary and human breast tissue.

**Radiation induces hyperplasia in mice lacking stromal PTEN.**
Based on our previous studies, stromal PTEN deletion alone is

insufficient to induce mammary epithelial transformation in the absence of an oncogene[3]. Therefore, we tested whether loss of stromal PTEN alone might sensitize mammary glands to radiation-induced epithelial cellular changes through paracrine EGFR/ErbB2 signaling. To this end, control (*Pten^{fl/fl}*) and experimental (*Fsp-cre;Pten^{fl/fl}*) mice were pre-treated with or without erlotinib for six days and were then exposed to a single dose of radiation (6 Gy whole-body). Mammary tissue was then transplanted into syngeneic wild-type recipients in order to mitigate secondary effects of whole-body radiation. After ~10 months, recipient mice were euthanized and transplanted mammary tissue was collected (Fig. 6a). Histological examination of the transplanted, irradiated mammary glands from *Fsp-cre; Pten^{fl/fl}* females treated with vehicle developed a significant (*p* < 0.001 by two-tailed Mann–Whitney) increase in lobuloalveolar hyperplasia, while control mammary glands from *Pten^{fl/fl}* females

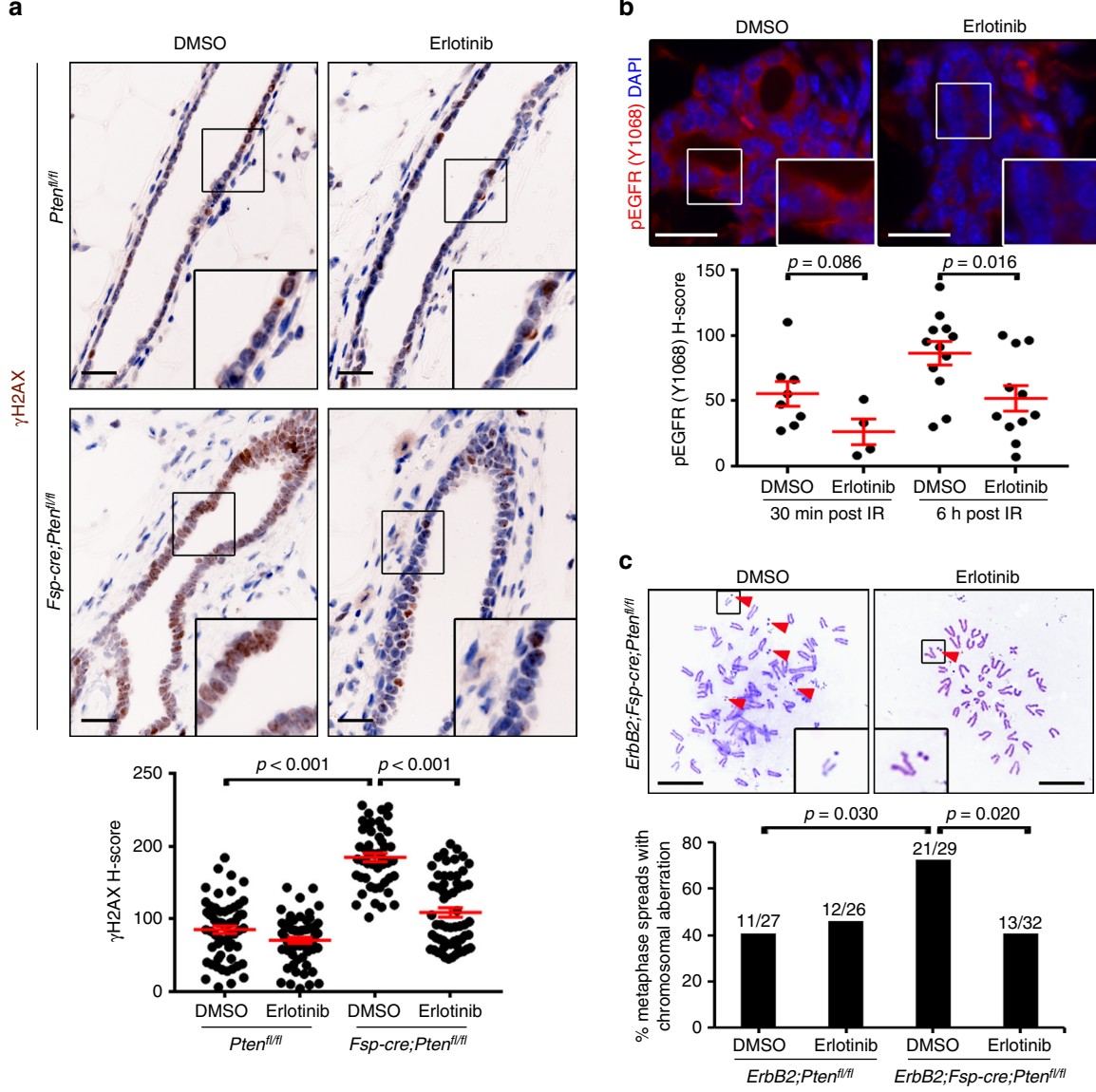

**Fig. 4** EGFR inhibition abrogates radiation-induced double-strand DNA breaks and genomic instability in stromal PTEN-null associated epithelium. **a** Representative γ-H2AX immunohistochemistry and quantification (mean ± s.e.m.) of mammary epithelium in *Pten^fl/fl* and *Fsp-cre;Pten^fl/fl* mice pre-treated with DMSO or erlotinib, irradiated (6 Gy whole-body) and evaluated 6 h post-radiation. DMSO (*Pten^fl/fl*) v. DMSO (*Fsp-cre;Pten^fl/fl*) *p* value determined by unpaired, two-tailed Student's *t* test. *Fsp-cre;Pten^fl/fl* DMSO v. erlotinib *p* value determined by two-tailed Mann-Whitney (*Pten^fl/fl*: DMSO *n* = 60 fields/6 mice, erlotinib *n* = 60 fields/6 mice; *Fsp-cre;Pten^fl/fl*: DMSO *n* = 48 fields/4 mice, erlotinib *n* = 60 fields/6 mice). Scale bar = 20 μm. **b** Representative phospho-EGFR (Y1068) immunofluorescence and quantification (mean ± s.e.m.) of mammary epithelium in *Pten^fl/fl* mice pre-treated with DMSO or erlotinib, irradiated (6 Gy whole-body) and evaluated 30 min or 6 h post-radiation (images are 6 h post IR). Both *p* values determined by unpaired, two-tailed Student's *t* test (30 min: DMSO *n* = 8 fields/2 mice, erlotinib *n* = 4 fields/1 mouse; 6 h: DMSO *n* = 12 fields/3 mice, erlotinib *n* = 11 fields/3 mice). Scale bar = 20 μm. **c** Representative metaphase spreads and quantification of chromosomal aberrations in epithelial cells of *ErbB2;Pten^fl/fl* and *ErbB2;Fsp-cre;Pten^fl/fl* mice pre-treated with DMSO or erlotinib, irradiated (6 Gy whole-body) and evaluated 1 week post-radiation in vitro. Arrowheads = chromosome break. Both *p* values determined by two-tailed Fisher's exact test (*ErbB2;Pten^fl/fl*: DMSO *n* = 27 spreads/2 mice, erlotinib *n* = 26 spreads/3 mice; *ErbB2;Fsp-cre;Pten^fl/fl*: DMSO *n* = 29 spreads/3 mice, erlotinib *n* = 32 spreads/3 mice). Scale bar = 10 μm

were completely normal (Fig. 6b, c; Supplementary Fig. 8a; Supplementary Table 3). Increased hyperplasia in irradiated stromal PTEN-null glands was not simply due to the deletion of stromal PTEN, as tissues not irradiated only exhibited non-significant alterations (Fig. 6b), consistent with our previous observations[3]. Importantly, erlotinib pre-treatment significantly (*p* = 0.003 by two-tailed Mann–Whitney) reduced hyperplastic lesions in irradiated *Fsp-cre;Pten^fl/fl* mammary tissue (Supplementary Table 3; Fig. 6b, c). Immunohistochemistry confirmed fibroblast-specific PTEN deletion in these tissues, where the

epithelium maintained PTEN (Fig. 6c; Supplementary Fig. 8a). The Rosa-lox-stop-lox-lacZ reporter[3,13] was further utilized to confirm *Fsp-cre* expression solely within stromal cell types indicating lack of EMT (Supplementary Fig. 8b).

## Discussion

The TME stroma is integral during all phases of tumor progression and metastasis[1,2], and our previous findings revealed the breast stroma could be tumor suppressive[3]. Our current study

aimed to determine whether stromal PTEN specifically suppresses tumor initiation. By analyzing pre-neoplastic mammary glands, we found that stromal PTEN is essential for maintaining genomic stability in adjacent epithelium. Specifically, stromal PTEN deletion upregulates fibroblast-specific EGF ligand expression and subsequent paracrine activation of adjacent epithelial EGFR/ErbB2 signaling. To the best of our knowledge, this is the first example where a stromal factor has been implicated in protecting the genome in a cell non-autonomous fashion.

Hyperactivation of both HER2/ErbB2 and EGFR has been previously linked to genomic instability[24–27], and our previous work has demonstrated that loss of stromal PTEN synergizes with the HER2/ErbB2 oncogene to hasten tumor formation[3]. Herein, we observed an increase in both ErbB2 and EGFR activity in the normal mammary epithelium associated with stromal PTEN deletion. These findings prompted us to utilize selective EGFR (erlotinib) and HER2/ErbB2 (CP-724,714) small molecule tyrosine kinase inhibitors to directly assess the functional

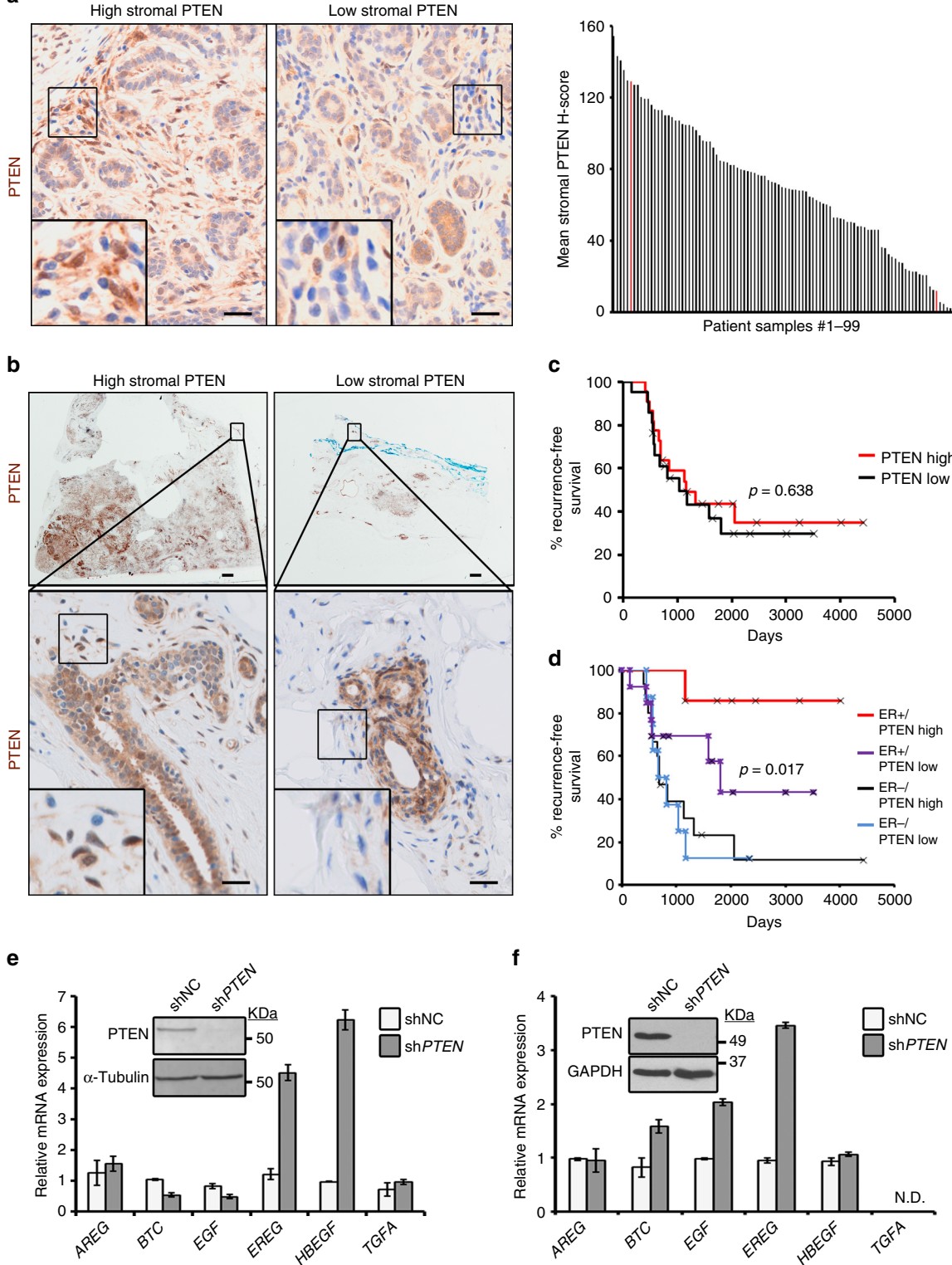

contribution of EGFR/ErbB2 signaling in the epithelial DNA damage response invoked by loss of stromal PTEN. Importantly, both inhibitors abrogated the increased DNA damage seen in mammary glands with stromal PTEN deletion as a response to radiation. In evaluating treatment efficacy, we observed a reduction in epithelial EGFR activity with both compounds, while ErbB2 was inhibited by CP-724,714, not erlotinib. These data, combined with the selective increase in EGF-specific ligands in PTEN-null fibroblasts, strongly support direct paracrine activation of epithelial EGFR as the primary signaling nexus resulting in the observed genomic instability; however, it is certainly likely that both EGFR/EGFR homo- and EGFR/ErbB2 hetero-dimers are involved given co-expression[29] and the known importance of dimerization in ErbB signaling[28]. While beyond the scope of our current work, we also cannot rule out the possible involvement of the other ErbB family members, ErbB3 and ErbB4.

It is crucial to acknowledge that while our work primarily focused on normal mammary tissue and early transformation, our findings also have functional implications in women with established breast cancer. Herein, we provide evidence that stromal PTEN levels in tumor-adjacent normal tissue is predictive of recurrence in a subset of HER2-positive patients with ER-positive disease. This patient cohort was carefully chosen to translate our mouse model (i.e. HER2-positive and treated with radiation therapy), but it is important to note that our data suggest that any DNA damaging agent, including many chemotherapeutics, could similarly lead to genomic instability in normal epithelial tissue associated with a PTEN-null stroma. Moreover, a PTEN-null stromal microenvironment likely has similar pro-tumorigenic effects upon residual EGFR/HER2-positive cancer cells from all tissue types that are not eliminated by first-line therapies. Interestingly, nuclear EGFR signaling has been shown to promote DNA repair in cancer cell line models, and through this mechanism, it confers radio-resistance[43–47]. While we did not observe nuclear localization of phospho-EGFR in the normal mammary epithelium, we predict that EGFR has distinct functions in normal versus transformed cells. Lastly, in a number of cancer types, EGFR inhibition radio-sensitizes tumor tissue through reduced DNA repair, as discussed above, as well as diminishes proliferation and hypoxia[46]. Thus, EGFR inhibition represents a unique treatment strategy as blockade should inhibit further transformation of normal cells, while preferentially killing tumor tissue. A deeper understanding of this mechanistic interplay in both tumor initiation and established disease is warranted.

While loss of stromal PTEN alone is insufficient to invoke invasive ductal carcinoma, the results described herein confirm that a single genetic change in the mammary stroma can elicit genomic instability in associated epithelium that upon a second carcinogenic hit (e.g. radiation) results in cellular changes and lobuloalveolar hyperplasia. Our additional studies indicate that these experimental findings have clinical implications for all women subject to chest radiation. Radiation is a critical treatment component for both solid and hematological malignancies, but it also increases the risk of secondary primaries, including breast cancer[38,48,49]. In breast cancer treatment, radiation increases cancer risk in the contralateral breast[50,51], and pediatric female patients treated with chest radiation for Hodgkin's lymphoma have a dramatically increased risk for developing aggressive breast cancer[52–57], with an estimated 35% developing bilateral breast cancer at a much younger age compared to the general population[52,55,58]. Furthermore, breast cancer risk is increased for patients historically treated with radiation for tuberculosis[59,60], scoliosis[61], mastitis[62], and benign breast disease[63]. Our current study provides clues to the etiologic mechanism for radiation-induced carcinogenesis. Given that we observed focal loss of PTEN in normal human breast tissue, our findings indicate that a percentage of patients with stromal loss of PTEN might be at risk for subsequent breast cancer via epithelial EGFR/ErbB2 activation and subsequent genomic instability. Notably, upon exposure to a single 6 Gy dose of whole-body radiation, which is substantially lower than the combined ~45–50 Gy used for localized breast cancer radiotherapy, stromal PTEN-null associated epithelium develops hyperplasia that can be completely prevented with EGFR inhibition. It is important to mention that whole-body irradiation is known to cause deleterious effects on hematopoietic, gastrointestinal and neurovascular systems[64,65], and while the increase in DNA damage observed in PTEN-null associated whole tissue mammary epithelium is similarly observed on epithelial cells treated with radiation ex vivo (Fig. 1f, g), we still must acknowledge the possibility of radiation-induced secondary effects on the mammary gland in the whole animal system.

Combined, this study provides substantial evidence that a subset of women who harbor low mammary stromal PTEN may be predisposed to developing radiation-induced breast cancer. Further investigations are warranted to determine if stromal PTEN could be utilized as a biomarker for radiation response, and in those patients exhibiting low stromal PTEN, pre-treatment with an EGFR inhibitor prior to radiation could possibly spare normal breast tissue from radiation-induced genomic instability and subsequent transformation.

## Methods

**Transgenic mice and genotyping**. Animal use was in compliance with federal and University Laboratory Animal Resources (ULAR) regulations under protocols 2007A0120-R3 (PIs-MCO and GMS) and 2007A0239-R2 (PI-GL) approved by the OSU Institutional Animal Care and Use Committee (IACUC). $Pten^{fl/fl}$ and $Fsp$-$cre$ mice were used[3,13]. $Fsp$ expression has been confirmed in the mammary stromal fibroblasts specifically and is undetectable in the epithelium, macrophages and endothelium[3,13]. $MMTV$-$ErbB2$ mice were generously provided by Dr. William Muller[5]. Wild-type FVB/N were commercially obtained from Jackson Laboratories (Bar Harbor, ME, USA) or Taconic Biosciences, Inc. (Cambridge City, IN, USA).

**Fig. 5** Low stromal PTEN is observed in normal breast tissue and predicts outcome in a HER2-positive breast cancer patient cohort. **a** Representative PTEN immunohistochemistry and quantification (mean of 4–10 representative fields/sample) in normal breast tissue isolated from women who underwent reduction mammoplasty ($n = 99$). Scale bars = 20 μm. **b** Representative PTEN immunohistochemistry in breast tumor samples isolated from women with HER2-positive disease treated with radiation therapy. Zoomed images show adjacent normal ductal tissue with either high or low stromal PTEN (high, $n = 22$ patients; low, PTEN $n = 21$ patients). Scale bars: top images = 1 mm; bottom images = 20 μm. **c** Kaplan–Meier analysis exhibiting recurrence probability within the HER2-positive patient population represented in (b) stratified by high versus low stromal PTEN (high, $n = 22$; low, $n = 21$). P value determined by Log-rank (Cox–Mantel). **d** Kaplan–Meier analysis exhibiting recurrence probability within the HER2-positive patient population represented in (b) stratified by high versus low stromal PTEN and ER status (ER-positive/PTEN-high, $n = 7$; ER-positive/PTEN-low, $n = 13$; ER-negative/PTEN-high, $n = 15$; ER-negative/PTEN-low, $n = 8$). P value determined by Log-rank (Cox–Mantel). **e** EGF ligand mRNA expression in primary cancer associated fibroblasts (CAFs) isolated from a breast cancer patient with and without PTEN lentiviral knockdown. Bars represent mean expression of technical replicates relative to $Gapdh$ ± s.e.m. Inset shows western blot for PTEN and α-Tubulin. **f** EGF ligand mRNA expression in primary normal human breast fibroblasts (>10 cm from tumor) with and without PTEN lentiviral knockdown. Bars represent mean expression of technical replicates relative to $Gapdh$ ± s.e.m. $TGFA$ is not detectable (N.D.). Inset shows western blot for PTEN and GAPDH

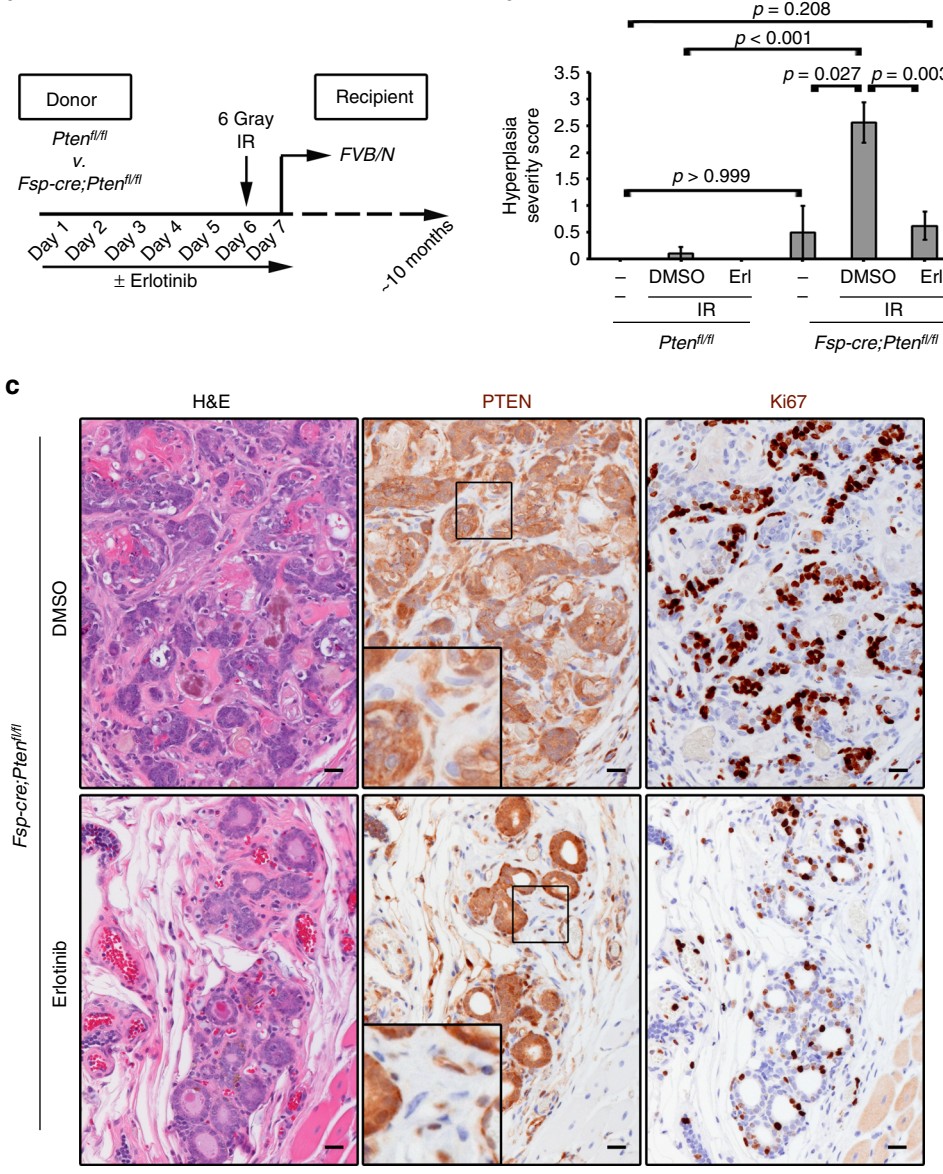

**Fig. 6** A single dose of radiation causes mammary hyperplasia in stromal PTEN-null mammary tissue and this effect that is abrogated by EGFR inhibition. **a** Donor (*Pten*$^{fl/fl}$ or *Fsp-cre;Pten*$^{fl/fl}$) mice were treated daily with DMSO or erlotinib for 7 days. On day 6, the mice were irradiated (6 Gy whole-body). On day 7, after their final dose with erlotinib, the mammary glands were transplanted into wild-type FVB/N recipients. The mammary glands were harvested ~10 months post-radiation and evaluated histologically. **b** Quantification of hyperplasia at 10 months post-transplant in *Pten*$^{fl/fl}$ or *Fsp-cre;Pten*$^{fl/fl}$ mammary tissue that came from a donor mouse that was un-manipulated (no drug and no IR) or that was pre-treated (± erlotinib) and irradiated as shown in (**a**) (*Pten*$^{fl/fl}$: No IR $n = 4$, DMSO $n = 9$, Erlotinib $n = 9$; *Fsp-cre;Pten*$^{fl/fl}$: No IR $n = 4$, DMSO $n = 9$, erlotinib $n = 8$). P values determined by two-tailed Mann–Whitney. **c** PTEN and Ki67 immunohistochemistry and associated H&E in *Fsp-cre;Pten*$^{fl/fl}$ mammary tissue in mice pre-treated with DMSO or erlotinib, irradiated (6 Gy whole-body), transplanted and evaluated 10 months post-radiation. Tissue pre-treated with DMSO displays lobuloalveolar hyperplasia with atypia. Scale bar = 20 μm

All experimental animals, obtained at a Mendelian ratio of 1 in 16, were virgin females 8–16 weeks of age and at least tenth-generation congenic (N10) FVB/N. Genotyping primer sequences are listed in Supplementary Table 4.

**Human subjects**. The normal reduction mammoplasty cohort has been previously discussed[34–36]. All patients comprising the HER2-positive patient cohort (OSUCCC) were deceased at the time of sample analysis excluding the need for IRB approval. All HIPAA policies remained in effect. Primary tumor blocks were reviewed (GT) and excluded if (1) the patient underwent neoadjuvant chemotherapy, (2) the sample lacked observable carcinoma, (3) the sample was not sizable (only core biopsy), (4) the sample had no adjacent normal, or (5) the estrogen receptor-α (ER) status was unknown. Human breast fibroblast primary

cultures were isolated under IRB protocol #1999C0262 (PI-LDY). Informed consent was obtained from each patient prior to isolation.

**Primary mammary fibroblast and epithelial cultures**. Mouse mammary (Pten$^{fl/fl}$ (wild-type), *Fsp-cre;Pten*$^{fl/fl}$ (PTEN-null or PTEN -/-), *ErbB2;Pten*$^{fl/fl}$, *ErbB2;Fsp-cre;Pten*$^{fl/fl}$ mice—all female FVB/N at 8–12 weeks of age) tissue was minced and digested with an enzymatic solution [0.15% Collagenase I (#C0130, Sigma, Saint Louis, MO, USA), 160U/ml Hyaluronidase (#H3506, Sigma), 1 μg ml$^{-1}$ hydrocortisone (#H4001, Sigma), 10 μg ml$^{-1}$ insulin (#I5500, Sigma) and 1% penicillin/streptomycin] in 5% $CO_2$ overnight at 37 °C. Tissue was neutralized with 10% FBS, 1% penicillin/streptomycin-DMEM (Thermo Fisher, Waltham, MA, USA), centrifuged and resuspended in 10% FBS, 1% penicillin/streptomycin-DMEM. Gravity separation was then utilized to separate out heavier epithelial organoids from the

fibroblast fraction. After 24 h, the epithelial culture media was replaced with DMEM/F12 (Thermo Fisher) supplemented with 5% horse serum, MEGM SingleQuot Kit (#CC-4136, Lonza, Basel, Switzerland) (bovine pancreatic extract (BPE), insulin, hydrocortisone, gentamycin, epidermal growth factor (hEGF), choleratoxin (0.02 ng ml$^{-1}$) (Sigma) and 0.06 mM calcium chloride); the fibroblast culture media was replaced with 10% FBS, 1% penicillin/streptomycin-DMEM. MMFs were spontaneously immortalized by passaging routinely (1–2 times a week) and seeding at the same density at each passage.

Human breast tissue was minced and digested with an enzymatic solution (2 mg ml$^{-1}$ Collagenase III, 1% penicillin/streptomycin, 10% FBS in DMEM) in 5% CO$_2$ overnight at 37 °C. Tissue was neutralized with 10% FBS, 1% penicillin/streptomycin$^-$DMEM, and gravity separation was utilized to separate out heavier epithelial organoids from the fibroblast fraction. Fibroblasts were plated and maintained in 10%-FBS, 1%-penicillin/streptomycin, 5ug ml$^{-1}$ plasmocin (InvivoGen, San Diego, CA, USA)-DMEM. After cells reached ~80% confluence, the human primary normal mammary fibroblasts were immortalized by transduction with hTERT lentivirus (pBABE-Hygro-hTERT #1773, Addgene, Cambridge, MA, USA) and selected for 10 days with 10 µg ml$^{-1}$ hygromycin. Both human primary normal and CAFs were transduced with pGIPZ control or shPTEN (clone ID V2LHS_231477) lentiviral constructs (GE Dharmacon, Lafayette, CO, USA) and selected for 12 days with 0.75 µg ml$^{-1}$ puromycin. None of these primary lines were tested for mycoplasma.

**Fluorescence activated cell sorting and microarray analysis.** Mammary tissue was dissociated with a McIlwain tissue chopper (Mickle Laboratory Engineering, Guildford, Surrey, United Kingdom) and incubated in Epicult-B medium (Stem Cell Technologies, Vancouver, British Columbia, Canada) supplemented with 5% FBS, 300U ml$^{-1}$ collagenase (Sigma) and 100U ml$^{-1}$ hyaluronidase (Sigma) for 1 h at 37 °C with gentle shaking. Red blood cells were lysed in ammonium chloride (Stem Cell Technologies). The digested tissue was then sequentially dissociated in 0.25% trypsin–EDTA (Sigma) for 1 min, 5 mg ml$^{-1}$ Dispase (Stem Cell Technologies) plus 0.1 mg ml$^{-1}$ DNase1 (Sigma) for 1 min and filtered through a 40 µm cell strainer. Lineage negative (Lin$^-$) subpopulations were then obtained through removal of CD45$^+$, Ter119$^+$, CD31$^+$ and BP-1$^+$ cells using the EasySep Mammary Epithelial Cell Enrichment kit as per manufacturer's instructions (Stem Cell Technologies). Lin$^-$ cells were further stained for CD24-PE (#553262, BD Pharmingen, San Jose, CA, USA), CD29-FITC (#555005, BD Pharmingen) and CD61-APC (#MCD6105; Invitrogen, Grand Island, NY, USA)[11,66,67]. Isotype controls for each antibody were used: PE-Rat IgG2b, κ isotype (#555848, BD Pharmingen); FITC-Hamster IgM, λ1 isotype (#553960, BD Pharmingen); APC-Hamster IgG, (#17-488-81, eBiosciences, San Diego, CA, USA). All procedures were performed using FACSAria (BD Biosciences, San Jose, CA, USA). Gating excluded cells labeled with isoform-matched control antibodies. If isotype-control was unclear, experiment was excluded. Viable cells were determined by DAPI exclusion. All data analyses were performed using FlowJo single cell analysis software.

For RNA analysis, each biological replicate of Lin$^-$CD24$^+$CD29$^{lo}$CD61$^-$mature luminal epithelial subpopulations was collected by pooling pre-neoplastic mammary tissue from 6 to 8 ErbB2;Pten$^{fl/fl}$ or ErbB2;Fsp-cre;Pten$^{fl/fl}$ female FVB/N mice at 8–10 weeks of age. This experiment was repeated five times to generate three independent samples per genotype of sufficient RNA quality for gene expression analysis. Total RNA was isolated using TRIzol (Invitrogen) and treated with DNAse I (DNA-free, Ambion). RNA quality was assessed using a NanoDrop TM2000 Spectrophotometer (Thermo Scientific). Sample quality to determine the best three of five samples was tested by characterizing size distribution via fluorescent capillary electrophoresis using the Bioanalyzer 2100 RNA 6000 Pico-Chip and Degradometer software (Agilent Technologies, Santa Clara, CA, USA). Expression was then evaluated using the GeneChip® Mouse Transcriptome Assay 1.0 (Affymetrix, Santa Clara, CA, USA) at the Genomics Shared Resource within the OSUCCC. Data were evaluated for statistical quality control and normalized in Expression Console (Affymetrix) using an SST RMA algorithm (GEO Accession #GSE93784) (https://www.ncbi.nlm.nih.gov/geo/query/acc.cgi?acc=GSE93784). GSEA v2.0 determined the enrichment of the Molecular Signatures Database (MSigDB) curated C5 gene sets within the data[68]. Statistical significance ($q < 0.05$) was determined through generation of a normalized enrichment score (NES) and a FDR[68].

**Mammary fat pad injection and tumor-free survival analysis.** Syngeneic primary mouse epithelial cells were isolated from the mammary glands of donor ErbB2;Pten$^{fl/fl}$ (round 1, n = 3; round 2, n = 4) and ErbB2;Fsp-cre;Pten$^{fl/fl}$ (round 1, n = 4; round 2, n = 3) female FVB/N mice at 8–9 weeks of age by Lin$^-$CD24$^+$/CD29$^+$ FACS as described above. Female FVB/N recipients at 8 weeks of age (total n = 8 per donor genotype) were anesthetized with isoflurane (Abbott Laboratories, Chicago, IL, USA). The inguinal nipple regions were swabbed with 70% ethanol. Four fat pads were injected per genotype with $1 \times 10^5$ cells in 50 µl 1× PBS, and the experiment repeated twice. All mice were euthanized at 338 days post-injection or at early removal criteria (ERC) as per IACUC regulations. Kaplan–Meier tumor-free survival curves were generated and statistical significance was determined using log-rank. Tumor volume was calculated by caliper measurements ex vivo [volume $= 1/2 \times$ length x (width)$^2$] at time of harvest.

**Whole-body radiation and drug treatments.** Eight to twelve week old female Pten$^{fl/fl}$, Fsp-cre;Pten$^{fl/fl}$, ErbB2;Pten$^{fl/fl}$ and ErbB2;Fsp-cre;Pten$^{fl/fl}$ FVB/N mice received whole-body 6 Gy radiation in the absence of anesthesia (RS2000 X-ray Irradiator; Rad Source, Suwanee, GA, USA). Animals were sacrificed and mammary tissue harvested at 30 min, 6 h or 1 week post-radiation. A subset of irradiated animals received prior treatment with the EGFR inhibitor erlotinib (Tarceva®, #E-4007, LC Laboratories, Woburn, MA, USA) or the ErbB2/HER2 inhibitor CP-724,714 (#A10242, AdooQ BioScience, Irvine, CA, USA). Erlotinib was resuspended in DMSO at 10 mg ml$^{-1}$ and mice received 50 mg kg$^{-1}$ or an equivalent volume of DMSO by oral gavage daily for 1 week. CP-724,714 was resuspended in 0.5% methycellulose (Sigma) at 10 mg ml$^{-1}$ and mice received 50 mg kg$^{-1}$ or an equivalent volume of 0.5% methycellulose by oral gavage daily for 1 week.

**Radiation and immunofluorescence on cultured cells.** For Rad51/keratin-8 (CK8) and γ-H2AX detection, epithelium was harvested from untreated ErbB2; Pten$^{fl/fl}$ and ErbB2;Fsp-cre;Pten$^{fl/fl}$ female FVB/N mice at 11–12 weeks of age by gravity separation as described above and cultured in 8-well chamber slides (Thermo Fisher). The next morning, cells were irradiated (3 Gy) (RS2000 X-ray Irradiator). At 6 h post-radiation, cells were fixed with 4% paraformaldehyde (FUJIFILM Wako Pure Chemical Corp., Tokyo, Japan), for 10 min, washed with 1× PBS and stored at 4 °C until stained. For pericentrin/α-tubulin detection, mice were treated with ± erlotinib and irradiated (6 Gy whole-body) followed by a 1 week recovery. Epithelial cells were then harvested by gravity separation as described above and cultured in 8-well chamber slides (Thermo Fisher). After reaching 70% confluency, cells were fixed with ice-cold methanol for 10 min, washed with 1xPBS and stored in DMEM (Thermo Fisher) without additives at 4 °C until stained. For PTEN/CK8 dual IF detection in the mature luminal population, cells were isolated by Lin$^-$CD24$^+$CD29$^{lo}$CD61$^-$ FACS as above and 40,000 viable cells (trypan blue exlusion (Sigma)) were cytospun on 13 mm single ringed cytology slides (Fisherbrand™) using single cytology funnels (Fisherbrand™). Cells were immediately fixed with 4% paraformaldehyde and stored in 1× PBS at 4 °C until stained.

For Rad51/CK8 and γ-H2AX staining, cells were washed with NET-Gel buffer, permeabilized with 0.2% Triton X-100 in 1× PBS for 10 min at room temperature (RT), and blocked with 5% bovine serum albumin (BSA) in 1× PBS for at least 30 min at RT before incubating overnight at 4 °C with primary antibodies in 1% BSA/PBS: rabbit anti-Rad51 (1:200, #NBP1-95892, Novus Biologicals, Littleton, CO, USA), rat anti-CK8 (1:400, Troma-I, Developmental Studies Hybridoma Bank, University of Iowa, Iowa City, IA, USA), mouse anti-γ-H2AX (Ser139) (1:5000, #05–636, EMD Millipore, Billerica, MA, USA), rabbit anti-pericentrin (1:250, #ab4448, Abcam, Cambridge, MA, USA) and mouse anti-α-tubulin (1:1000, #T6199, Sigma). Cells were then washed with 1× PBS, probed with Alexa Fluor® 488 and 594 conjugated secondary antibodies (Thermo Fisher) in 1% BSA/PBS, washed and mounted with SlowFade® Gold antifade reagent with DAPI (#S36938, Thermo Fisher). For PTEN/CK8 staining, cells were washed with 1× PBS, permeabilized in 0.5% Triton X-100 in 1× PBS for 10 min at RT, and blocked with 5% donkey serum in 0.3% Triton X-100/1 x PBS for 1 h at RT before incubating overnight at 4 °C with primary antibodies in blocking solution: rabbit anti-PTEN (1:200, #9559, Cell Signaling) and rat anti-CK8 (1:400). Cells were then washed with 1× PBS, probed with Alexa Fluor® 488 and 594 conjugated secondary antibodies in 1× PBS, washed and mounted with SlowFade® Gold antifade reagent with DAPI.

Imaging for Rad51, pericentrin/α-tubulin and PTEN/K8 was done using the Eclipse E800 microscope (Nikon Instruments Inc., Melville, NY, USA) using the MetaVue™ Research Imaging system (Molecular Devices, Sunnyvale, CA, USA). Imaging for γ-H2AX was done using the VECTRA® Automated Quantitative Pathology Imaging system (PerkinElmer, Hopkinton, MA, USA).

**Immunofluorescence on paraffin sections.** Tissue sections were baked at 65 °C for 15 min and de-paraffinized using a xylene-ethanol series. This was followed by antigen retrieval using EDTA Decloaker 5 × (BioCare, Pacheco, CA, USA) in a steamer for 40 min. The sections were then cooled to RT, blocked for 1 h at RT with 5% BSA/0.5% Tween-20 in 1× PBS, and then incubated with rabbit anti-phospho-EGFR (Tyr1068) (1:25, #2234, Cell Signaling, Danvers, MA, USA) or rabbit anti-phospho-HER2/ErbB2 (Tyr1221/22) (1:25, #2243, Cell Signaling) in blocking buffer at 4 °C overnight. The slides were then washed thrice in 1× PBS and incubated with Alexa Fluor® 594 conjugated secondary diluted in 1× PBS (1:250) in the dark for 1 h. Following another set of PBS washes, the slides were mounted in Slowfade® Gold Antifade with DAPI (Thermo Fisher) and stored at 4 °C in the dark. Imaging was done using the VECTRA® Automated Quantitative Pathology Imaging system (PerkinElmer, Hopkinton, MA, USA).

**Immunohistochemistry on paraffin sections.** Sections were stained using the Bond RX autostainer (Leica Biosystems Inc., Buffalo Grove, IL, USA). Briefly, slides were baked at 65 °C for 15 min and the automated system performed dewaxing, rehydration, antigen retrieval, blocking, primary antibody incubation, post primary antibody incubation, detection (DAB), and counterstaining using Bond reagents (Leica). Samples were then removed from the machine, dehydrated through a series of ethanol and xylenes and mounted. Rabbit anti-γ-H2AX (Ser139) (1:800, #9718, Cell Signaling), rabbit anti-phospho-EGFR (Tyr1068) (1:50, #2234, Cell Signaling),

rabbit anti-phospho-HER2/ErbB2 (Tyr1221/22) (1:400, #2243, Cell Signaling), rabbit anti-PTEN (1:150, #9559, Cell Signaling: used for mouse transplants and normal reduction mammoplasty samples), mouse anti-PTEN (1:100, #M3627, Dako: used for HER2 breast cancer patient cohort), rabbit anti-Ki67 (1:200, #ab16667, Abcam) were diluted in antibody diluent (Leica). TUNEL staining was performed using manufacturer's recommendations (ApopTag Peroxidase In Situ Apoptosis Detection Kit, #S7100; EMD Millipore). All microscopic imaging was done using the VECTRA® Automated Quantitative Pathology Imaging system or an Axioskop 40 with ZEN Software (Zeiss, Germany). Whole tissue imaging was done using a Stemi SV 11 Stereoscope with ZEN Software (Zeiss).

**X-gal staining**. Frozen tissue sections were dried at RT, fixed in a glutaraldehyde solution (0.2% glutaraldehyde, 1.25 mM EGTA, pH 7.3 and 2 mM magnesium chloride in 1× PBS), washed with LacZ wash buffer [2 mM magnesium chloride, 0.01% sodium deoxycholate, 0.02% IGEPAL CA-630 (Sigma) in PBS] and then stained overnight at 37 °C protected from light in LacZ staining solution (4 mM potassium ferricyanide, 4 mM potassium ferrocyanide, 1 mg ml⁻¹ X-gal in LacZ wash buffer). Stained sections were then washed with 1× PBS and rinsed with water before counterstaining with nuclear fast red. Microscopic imaging was performed using an Axioskop 40 with ZEN Software (Zeiss). Whole tissue imaging was done using a Stemi SV 11 Stereoscope with ZEN Software (Zeiss).

**Quantification of immunostaining**. Quantification of PTEN/CK8 dual corrected total cell fluorescence was performed in Image J[69] by manually selecting all the cells within the 40× field (circle tool) and measuring the area and integrated density for each. Corrected total cell fluorescence was determined for each cell (integrated density minus the cell area) and averaged across all cells per field (5 representative fields (40×) per genotype (3 pooled mice/genotype)).

Quantification of Rad51 on cultured cells was performed in Image J[69] by manually counting the number of Rad51 + cells along with determining total cell number by thresholding the DAPI channel and analyzing particles (2–5 representative fields (20–40×) per mouse (3 mice/*ErbB2;Pten*^fl/fl^, 2 mice/*ErbB2;Fsp-cre;Pten*^fl/fl^)).

Quantification of γ-H2AX (Ser139) on cultured cells was performed using the pattern recognition algorithm in the Inform® software (PerkinElmer). The spectrally unmixed Alexa Fluor® 594 signal was scored based on a user defined threshold into four categories (0+, 1+, 2+ and 3+). The percent of cells within each scoring category was determined based on cell segmentation with the DAPI counterstain (1–6 representative fields (20×) per mouse (3 mice/genotype)). An H-Score was then calculated using following formula: (1×(% cells 1+)+2×(% cells 2+)+3×(% cells 3+)).

Quantification of Pericentrin/γ-tubulin on cultured cells was performed by manually counting the number of centrosomes within all observable metaphase spreads in Image J[69] (5–16 spreads (40×) per mouse (2 mice/genotype)).

Quantification of phospho-ErbB2 (Tyr1221/22) IHC was performed by measuring the optical density (OD) of the DAB signal at a fixed threshold in NUANCE, which was then divided by the % epithelial area per field of view as determined by thresholding the hematoxylin channel and analyzing particles in Image J[69] (5–8 representative fields (40×) per mammary gland; 2 glands/mouse; 4 mice/genotype). Quantitation of phospho-ErbB2 (Tyr1221/22) IF was determined by measuring the area of the red (Alexa Fluor® 594) signal at a fixed threshold in ImageJ[69], which was then divided by the % epithelial area per field of view as determined by thresholding the blue (DAPI) signal and analyzing particles in Image J[69] (4–8 representative fields (40×) per mammary gland; 1–4 mice/genotype).

Quantification of epithelial γ-H2AX (5–18 representative fields (40×) per mammary gland;≥3 mice per genotype/treatment group], epithelial phospho-EGFR (Tyr1068) (IF: 3–6 representative fields (40×) per mammary gland; 2–3 mice/treatment group; IHC: 8 representative fields (40×) per mammary gland; 2 glands/mouse; 3 mice/genotype), epithelial Ki67 (3–6 representative fields (40×) per mammary gland; ≥3 mice per genotype), epithelial TUNEL (3–8 representative fields (40×) per mammary gland; ≥3 mice per genotype/treatment group), and stromal PTEN [99 reduction mammoplasty patient samples: 4–10 representative fields (40×) per sample; 43 HER2-positive breast cancer patient samples: 1 field (40×) per block depicting region with lowest stromal PTEN, 1–4 blocks per patient sample were averaged] were performed using the pattern recognition algorithm in the Inform® software (PerkinElmer). Briefly, each image underwent manual tissue segmentation to select only the epithelium (for γ-H2AX, Ki67, TUNEL and phospho-EGFR) or only the stroma (for PTEN). The spectrally unmixed DAB signal was scored based on a user defined threshold into four categories (0+, 1+, 2+ and 3+). The percent of cells within each scoring category was determined based on cell segmentation with the hematoxylin counterstain. An H-Score was then calculated using following formula: (1×(% cells 1+)+2 × (% cells 2+)+3 × (% cells 3+)).

All images in the main figures underwent final processing using Adobe Photoshop® and in the supplement using Microsoft PowerPoint for brightness/contrast/sharpness when appropriate. Identical adjustments were made over the entire image and for all images per experiment.

**Karyotyping**. *ErbB2;Pten*^fl/fl^ and *ErbB2;Fsp-cre;Pten*^fl/fl^ female FVB/N mice at 9–16 weeks of age were treated with ± erlotinib and irradiated (6 Gy whole-body) followed by a 1 week recovery. Epithelial cells were harvested by gravity separation as described above and cultured in 10 cm tissue culture dishes (Thermo Fisher). Once cells reached 70% confluency, 50 μl KaryoMAX® colcemid® (Gibco #15210-040) per 10 ml media was added, and the cells incubated at 37 °C for 2 h to arrest proliferating cells in M-phase. The media, 1× PBS wash and trypsinized cells were combined, neutralized and centrifuged. 10 ml of fresh pre-warmed 0.56% potassium chloride (KCl) solution was added to the cell pellet while avoiding cell clumping and incubated at 37 °C for 30 min. Cells were then fixed with 1 ml ice-cold 3:1 MeOH:Glacial acetic acid and centrifuged gently to remove the fixative-KCl solution. The addition of fixative and centrifugation was repeated twice, each with 5 ml of ice-cold fixative. Fixed cells were stored at −20 °C until metaphase spreads were made. Before making metaphase spreads, the fixation step was repeated at least two more times. Microscope slides were wiped clean with 70% EtOH to avoid dust. 30 μl of the fixed epithelial cell preparations were dropped onto the slide in two drops and the drop gently dried by blowing. Once dry, these metaphase spreads were stained with 1:10 KaryoMAX® Giemsa Stain Improved R66 Solution "Gurr" (Gibco #10092-013) for 1 min and washed thoroughly in water. Then slides were dried using a blow-dryer. Metaphase spreads were observed at 100× magnification. All readable mitotic figures were analyzed and DNA damage was quantified as the sum of the following: (i) counting the number of chromosomes to determine chromosome amplification or aneuploidy, (ii) counting single-strand or double-strand breaks to determine defects in DNA repair and (iii) estimating translocations. For the experiment in the absence of drug, 1–23 fields (100×) per mouse were quantified (4 mice per genotype). For the experiment ± erlotinib, 4–23 fields (100×) per mouse were quantified (2–3 mice per genotype: *ErbB2*-DMSO n = 2 and erlotinib n = 3; *ErbB2;Fsp-cre;Pten*^fl/fl^-DMSO n = 3 and erlotinib n = 3). Images were taken using the 100× lens on the Eclipse 50i microscope (Nikon) mounted with an Axiocam HRc camera (Zeiss).

**Genomic DNA analysis and quantitative real-time PCR**. To isolate genomic DNA from tumor tissue, formalin-fixed paraffin embedded (FFPE) sections were processed using the QIAamp DNA FFPE Tissue Kit (Qiagen, Hilden, Germany). Standard PCR reactions were carried out using primers for *Pten* floxed and deleted alleles as in Supplementary Table 4 and PCR products visualized by agarose gel electrophoresis. To isolate RNA from total tumor tissue, FFPE sections were processed using the FFPE RNA Purification Kit (Norgen Biotek Corp. Ontario, Canada). To isolate total RNA from cells, the TRIzol reagent was used according to manufacturer's instructions (Invitrogen). RNA was treated with DNAse I (DNA-free, Ambion, Grand Island, NY, USA), and cDNA generated via SuperScript III Reverse Transcriptase (Invitrogen). For tumor qRT-PCR analysis, the Universal Probe Library system was used (Roche, Applied Biosystems, Foster City, CA, USA). Sample quality was verified by comparing Ct values for mouse ribosomal gene *Rpl4*. Primer set-probe combinations are detailed in Supplementary Table 5.

For cDNA generated from human fibroblasts and FACS isolated mature luminal cells, a pre-amplification step was performed following the TaqMan® (Applied Biosystems, Grand Island, NY, USA) PreAmp Master Mix protocol as per manufacturer's instructions. qRT-PCR was performed using TaqMan® Gene Expression Assays. Assay IDs are listed in Supplementary Table 6.

**Immunoblotting and enzyme-linked immunosorbent assays**. For immunoblotting, fibroblasts were lysed on ice (50 mM Tris-HCl, pH7.4; 100 mM NaCl; 1 mM EDTA; 1 mM EGTA; 1 mM NaF; 0.1% SDS; 0.5% Sodium Deoxycholate; 1% Triton-X-100; 10% Glycerol; Protease and Phosphatase Inhibitor Cocktails (Sigma)), and protein was quantified (Bradford Assay, Bio-Rad, Hercules, CA, USA). Protein lysate was resolved using SDS-PAGE, and transferred to PVDF membrane (EMD Millipore). For human CAFs, the LiCOR Odyssey TBS Blocking Buffer (Lincoln, NE, USA) was used to block and as a diluent for both primary (PTEN (1:1000, Cell Signaling, #9559); α-tubulin (1:2000, Sigma, #T6199)) and secondary antibodies (LiCOR). Signal was detected using the LiCOR Odyssey®. For mouse and human normal fibroblasts, 5% milk-1xTBST was used to block and as a diluent for both primary (PTEN (1:1000, Cell Signaling, #9559; TGF-α (1:1000, Cell Signaling, #3175); GAPDH (1:2000, Santa Cruz, #sc-25778)) and secondary HRP-conjugated antibodies (GE Healthcare, Little Chalfont, UK). Signal was detected using the ECL detection kit (Thermo Fisher) and developed using X-ray film. All unprocessed, uncropped blots are included in Supplementary Figs. 9–11.

For ELISAs, equivalent numbers of viable WT control (*Pten*^fl/fl^) and PTEN-null (*Fsp-cre;Pten*^fl/fl^) MMF cells were seeded in complete 10% FBS-DMEM media. The next day, the media was changed to serum-free DMEM and media was conditioned overnight. Conditioned media was then isolated and concentrated 5–10 fold using Amicon Ultra-15 Centrifugal Filter Units (Ultracel-3, EMD Millipore). Epiregulin and amphiregulin protein levels were determined using mouse ELISAs (EMEREG and EMAREG, Thermo Fisher) as per manufacturer's recommendations. Final concentration was adjusted for the total cell number at time of harvest.

**Mammary tissue transplantation**. Recipient female mice (FVB/N, Taconic) at 8–9 weeks of age were anesthetized (isoflurane/oxygen) and the scapular region shaved. 24 h later, whole inguinal mammary tissue lacking associated lymph nodes

was harvested from donor females and placed subcutaneously into the scapular region of recipient females under aseptic conditions. Incisions were closed using a 9 mm wound clip, which was removed 10 days post-transplant. Recipient animals were monitored biweekly for ~10 months, at which the mice were sacrificed and mammary tissue harvested for histology.

**Pathological characterization of hyperplasia.** Transplanted tissue was characterized as having (1) hyperplasia, or (2) hyperplasia with atypia (MIN—carcinoma in situ). The glands were first classified based on the degree of development of the mammary gland tree as lobule type 1, 2, and 3[70]. The cutoff values for defining lobule type were set as follows: lobule type 1 = 2–15 ductules/alveoli per cross section; lobule type 2 = 15–30; and lobule type 3 = >30. Lobuloalveolar hyperplasia was defined as an increase in profiles of alveoli beyond lobule type 1 (lobule 2 and 3) without cytologic changes in alveolar or ductal cells and preservation of the lobular architecture. Lobulaveolar hyperplasia was further graded as minimal (lobule type 2 with focal distribution), mild (lobule type 2 with multifocal distribution) or moderate (lobule type 3 with multifocal to coalescing distribution).

Criteria for diagnosing hyperplasia with atypia (MIN—carcinoma in situ) included irregular proliferation of the ductal or alveolar epithelium, cellular atypia and/or pleomorphism and mitotic figures. The basement membrane remained intact. Finally, a pathological score was assigned to each tissue as follows: 0 = unremarkable (lobule 1); 1 = hyperplasia, minimal; 2 = hyperplasia, mild; 3 = hyperplasia, moderate; 4 = hyperplasia with atypia.

**Statistical methods.** Power analyses were not performed a priori. Experimental mice were obtained at a Mendelian ratio of 1 in 16 and were randomly utilized within genetic groups. All analyses and differences between distinct genetic groups were done in a blinded fashion. For all data, normality assumption was checked by D'Agostino–Pearson omnibus, Shapiro–Wilk and Kolmogorov–Smirnov normality testing. Data were considered normally distributed upon passing any of the three tests. For continuous data, ANOVA was used to determine differences among several groups, while t-test was used for pairwise comparisons (homoscedastic or heteroscedastic Student's t-test as appropriate). All analyses using Student's t-test were two-tailed. For data not normally distributed, statistical comparisons were done by two-tailed Mann–Whitney. Statistical significance for all contingencies tables was determined by two-tailed Fisher's exact. All charts depict the mean ± the standard error of the mean.

**Data availability.** Gene expression data supporting these findings can be found on the Gene Expression Omnibus (accession no. GSE93784). All other data are available within the article and its supplementary information or from the corresponding author upon reasonable request.

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

## Acknowledgements

The authors graciously thank Drs. Nicholas Denko and Terence Williams for critical review of the manuscript. The authors also thank Jason Bice and Daphne Bryant (Histology Core), Bryan McElwain and Katrina Miller (OSUCCC-Analytic Cytometry) and Sarah Warner (OSUCCC-Genomics) for technical support. This study was supported by the NIH (P01CA097189, MCO, GL and MP), the Department of Defense (W81XWH-14-1-0040, GMS), the Hollings Cancer Center Support Grant (P30 CA138313), and the Pelotonia Fellowship Program (GMS, SB and KAT).

## Author contributions

All authors assisted with data interpretation and manuscript review. G.M.S., S.B., A.M.H., S.T.S., G.L., and M.C.O. conceived and designed the research. G.M.S. performed TUNEL staining, imaging and quantification; pErbB2, pEGFR, and Ki67 IHC imaging and quantification; PTEN imaging and quantification in human cohorts; and qRT-PCR and western blot analysis of normal fibroblasts. G.M.S. and S.B. performed FACS, microarray analysis and GSEA. G.M.S., S.B., K.A.T., and S.A.S. performed in vivo γ-H2AX imaging and quantification. G.M.S., S.B., A.M.H., and S.A.S. performed drug treatments. G.M.S., S.B., and S.T.S. performed radiation. G.M.S., S.B., and A.J.T. performed mammary transplantation. G.M.S. and S.A.S. performed pEGFR and pErbB2 IF imaging and quantification. S.B. performed staining and subsequent imaging and quantification for in vitro γ-H2AX, RAD51, PTEN and pericentrin/α-tubulin IF; karyotyping-imaging and quantification. S.B. and D.P. performed karyotyping. KAT performed PTEN knockdown in human CAFs along with subsequent qRT-PCR and western analysis as well as tumor cohort quality control PCR/qRT-PCR. A.M.H. performed subsequent quantification for human normal breast tissue PTEN IHC and in vitro RAD51 IF. S.T.S. performed Kaplan–Meier analyses. A.J.T. performed X-gal staining and along with N.S. and S.A.S. maintained the mouse colony. M.C.C. and T.R. performed pathological review of mammary transplants. G.T. reviewed HER2 breast cancer sample blocks. R.D.K. performed all in vivo γ-H2AX, pErbB2, pEGFR, PTEN, Ki67 IHC. MD performed the ELISAs and qRT-PCR on erlotinib in vitro treatments. S.M. performed PTEN lentiviral knockdown in human normal breast fibroblasts. S.K. and P.G.S. provided human normal breast tissue. G.M.S., S.T.S., L.Y., S.A.F. performed all statistical analyses. S.T.S., A.C., and J.R.W. provided radiation oncology expertise. L.D.Y. provided human fibroblasts and breast surgical oncology expertise. T.L. provided DNA repair expertise. M.P. provided human TME expertise. G.M.S. wrote the manuscript and prepared figures. G.L. and M.C. O. supervised the research.

## Additional information

**Competing interests:** The authors declare no competing interests.

