## [Peer Review File · Nature Communications]

Reviewers' comments:

Reviewer #1 (Remarks to the Author):

In this manuscript, Sizemore et al. demonstrate that loss of PTEN induces transformation of the neighboring epithelial cells. They go on to show that PTEN loss in stroma results in genomic instability in the neighboring epithelial cells which could be tumor promoting. Additionally, the authors find that loss of PTEN in the stroma results in upregulation of EGF ligands and a corresponding increase in EGFR signaling in surrounding tissues. Suppression of EGFR signaling via an EGFR inhibitor is sufficient to block the observed increase in genomic instability in neighboring epithelial, directly demonstrating that the elevated EGFR signaling caused by stromal deletion of PTEN induces genomic instability in neighboring epithelial cells. These data, along with several interesting *in vivo* experiments detailed in this manuscript, strongly support the use of EGFR inhibitors during radiation treatments to reduce off-target effects and the risk of developing secondary malignancies.

This is a very thorough and important study to determine the contribution of PTEN null stroma on mammary tumorigenesis. The main conclusion of this study is based on the fact/assumption that PTEN is deleted only in the fibroblasts and not in the epithelial cells. Given the importance of these results and the clinical implications that this study brings forth, it is imperative that the authors completely rule out that there is no loss of PTEN in the epithelial population of ErbB2;Fsp-cre;Ptenfl/fl mice. This and other concerns/comments are listed below.

Major Points:

1. Authors have used fibroblast specific-1 promoter (Fsp1) for Cre expression - which should induce Cre expression specifically in the fibroblasts (as shown by authors in their previously published work). However, there is always a concern for slight 'leakiness' of a promoter which would mean that cells which shouldn't be expressing any Cre (eg. epithelial cells) - might do that, albeit at a very low level. Given that it only takes a small subset of transformed cells to induce tumorigenesis, it is critical that the authors confirm that there does not exist any such epithelial cell sub population wherein PTEN is lost or is expressed less than it should. PTEN haploinsufficiency is well documented so even a reduced level (instead of complete knockdown) of PTEN could be enough to induce tumor promoting effects. We recommend the following experiments (or a combination of them) to help demonstrate that these effects are driven solely by loss of PTEN in the stroma.

a) It would help if the authors FACS sort the epithelial cell population, and demonstrate via western blot that PTEN levels are not affected in the epithelial cells and also carry out PCR analysis of the floxed allele in the epithelial cells to show that there is absolutely no deletion of PTEN floxed alleles in these epithelial cells.

b) Instead of injecting the PTEN positive epithelial cells (from mice which had PTEN null stroma), it will be more powerful to take the PTEN null stroma, devoid of any epithelial cells, and then inject those in the wt mice. Others have used this approach successfully to study the effect of stroma on mammary tumorigenesis. If injecting PTEN null stroma induces tumor formation in wt mice - this will go a long way in confirming the conclusions put forth by the authors.

c) Another approach the authors could use is to transplant labeled (wt) epithelial cells in ErbB2;Fsp-cre;Ptenfl/fl and ErbB2;Ptenfl/fl mice; allow them to grow in the presence of PTEN-depleted stroma and then FACS sort these labeled cells and now inject these cells in wt mice (as done for Fig. 1b). If these cells form tumors in the recipient mice - there will be no doubt that these PTEN wt cells which grew in presence of PTEN null stroma are transformed. The authors can also look for signs of genomic instability in these 'labeled' wt epithelial cells that have grown in presence of Pten null stroma (vis-a-vis Fig 2c).

2. MMTV-ErbB2 is an established mouse model to study mammary tumorigenesis. There is a concern that in ErbB2;Fsp-cre;Ptenfl/fl mice, ErbB2 dependent tumorigenesis (irrespective of PTEN

null stroma) could induce EMT which would essentially mean that the PTEN floxed epithelial cells in ErbB2;Fsp-cre;Ptenfl/fl mice undergoing early EMT- associated changes will now start expressing Cre leading to loss of PTEN from the epithelial cells.

3. Fig. 1f and Fig.2: It has been well documented now (Loonstra, Jos Jonkers et al. PNAS 2001, Marc Schmidt-Supprian & Klaus Rajewsky, Nature Immunology, 2014) that expression of Cre is DNA damage inducing independent of its activity on floxed DNA. Authors show that only the epithelial cells from ErbB2;Fsp-cre;Ptenfl/fl mice show DNA damage and not those from ErbB2;Ptenfl/fl mice. However, this could also be because of Cre expression in stroma of ErbB2;Fsp-cre;Ptenfl/fl mice which in turn could result in high DNA damage in stroma/fibroblasts resulting in DNA damage induced signaling and effects on the surrounding epithelial cells. This could be PTEN loss independent. An important control missing here is stroma that is expressing Cre but does not have floxed PTEN. It is a better control than the ones used by the authors (no Cre but floxed Pten alleles). It will help to rule out that the DNA damage effect - including chromosomal abnormalities are not simply because of DNA damage induced by Cre recombinase. Another piece of information missing here is the status of DNA damage in the stromal cells. Is there increased DNA damage in the stroma of ErbB2;Fsp-cre;Ptenfl/fl mice as well? And what is the status of DNA damage in stromal cells which are expressing Cre but do not have the floxed PTEN?

4. Fig. 1f. What are the Rad51 protein levels by western blot. Is Rad51 downregulated as suggested by the GSEA analysis?

5. The authors show that the epithelial cells derived from the ErbB2;Fsp-cre;Ptenfl/fl mice but not the ErbB2;Ptenfl/fl mice give rise to tumors when transplanted into wt mice (Figure 1B) and have clear gene expression changes, suggesting that these cells have been primed for tumorigenesis by stroma depleted of Pten. Authors show that this is due to elevated EGFR signaling. To confirm this conclusion, it will help if the authors could determine in vitro if exogenous elevation of EGF signaling results in gene expression changes similar to those seen in Figure 1E, and will lead to similar DNA repair defects as predicted by their EGFR inhibitor assays.

6. Given that the authors' main observation (epithelial cell transplantation expt; Fig 1b) is based on experiments carried out with ErbB2;Fsp-cre;Ptenfl/fl and ErbB2;Ptenfl/fl mice, it is not clear why the authors show half of the genomic instability data in non-ErbB2 mice (Fsp-cre;Ptenfl/fl and Ptenfl/fl mice) - especially when studying the upregulation of EGF ligands. Is this upregulation happening in mammary epithelial cells of ErbB2;Fsp-cre;Ptenfl/fl mice compared to ErbB2;Ptenfl/fl mice. It is important to know that given that the tumor studies are all done in ErbB2 mouse model.

Reviewer #2 (Remarks to the Author):

The paper by Sizemore and colleagues shows how loss of stromal PTEN in mammary fibroblasts impacts in EGFR/ErbB2 activation in the mammary epithelium and on the response of these cells to radiotherapy. As models they use epithelium from the normal gland and the transgenic MMTV-ErbB2 gland. Stromal PTEN is deleted by using Fsp-cre PTEN deletion in which the tumor suppressor is lost in cells of mesenchymal origin. In the past they showed that if PTEN is deleted in the stromal cells of the ErbB2 transgenic gland this causes the accelerated appearance of invasive tumors (2009 Nature paper).

The work presented in this manuscript shows the novel result that loss of stromal PTEN causes upregulation of multiple EGF family ligands in stromal cells and that both EGFR and ErbB2 are phosphorylated in mammary epithelial cells. This has a strong impact on the DNA damage

response of the gland to radiotherapy. They also show that a dose of whole-body radiation leads to focal lobuloalveolar hyperplasia, which is prevented if the mice are treated with an EGFR inhibitor. This inhibitor also causes a decrease in the DNA repair IR essentially back to the levels seen in control glands with Pten. The results are novel and interesting and by bringing in analyses PTEN levels in HER2-positive human tumors, the work could in the future be clinically relevant.

I have some questions and suggestions below that if addressed would make the work even stronger.

Major comments:

1- in Fig 1 they look at effects of stromal loss of Pten on tumor formation and in the DNA repair response. Panels a and d show FACS analyses on cells from the mammary epithelium, which was analyzed then purified by CD24/CD29 positivity before being transferred into WT recipients. They mention that there are no differences in the populations. However, in a recent paper from the group (Sizemore et al Oncogene 2017), they show in Fig 2 that there is an expansion of the basal/myoepithelial population in the Pten null stroma. Please clarify. Could this be due to a difference in the time when the cells were harvested from the mammary glands?

2- in Fig 3 they show that in mammary glands that are Pten null not only EGFR is P, but also ErbB2 (panels c & d). This is not unexpected considering that EGF family ligands activate EGFR homo- and ErbB2-containing EGFR-heterodimers. In addition the NRGs also activate ErbB2 in ErbB receptor heterodimers.

In Fig 4 they use the ErbB2/Pten null model for some experiments and show that chromosome breaks are lower in the erlotinib treated mice (Panel c). In panel c they show that P-EGFR levels are also lower. The data in panel b seem to be from Pten null mammary epithelium – or is this also the ErbB2/Pten null model? What about P-ErbB2 levels? Are they also affected by the EGFR inhibitor erlotinib?

3- I think it would also be important to look at the impact of an ErbB2 inhibitor, in at least some of the experiments. This is particularly relevant since in the human breast cancer studies shown in Fig 5, they concentrate on patients with elevated HER-2/ErbB2. These patients will receive an inhibitor to ErbB2 and not to the EGFR. For experiments with their mouse models the ErbB2 inhibitor lapatinib would be perfect.

4- Fig 5. It is interesting that CAFs and normal fibroblasts taken from an area >10cm from the tumor do show an upregulation of EGF-family ligands in response to PTEN KD (panels e & f). The particular ligands are different though. What about NRG4 that was highly upregulated in Fig 3a – was it tested? Can they say anything about the mechanism downstream of PTEN loss?? What happens to the activity of MAPK and PI3K in these cells? Do the ligands share a TF binding site in their promoter that might be responsible for the transcriptional increase?

The same question can be asked in the mouse models. In the absence of PTEN is there an increase in stromal activity of these pathways?

Finally, out of curiosity – the data shown in Fig 5e in which PTEN was KDd in CAFs from a breast cancer patient – did this patient's stroma have high or low PTEN levels? Can they link overall PTEN levels this with any particular increase/or not in EGF-family ligands?

5- While it is very likely that downregulation of P- EGFR in the mammary glands results from the kinase blocking the receptor activity, one cannot rule out the possibility that ligand expression is also decreased in response to erlotinib. In addition, the authors have only looked at ligand RNA levels and have not measured actual protein levels. Protein levels of at least some of the ligands should be examined in the CM of the fibroblasts. In addition the response of the fibroblasts to

erlotinib treatment could be directly checked in the cultured MMFs used in Fig 3 and/or the CAFs and the fibroblasts used in Fig 5. For this latter experiment RNA levels of the ligands that they show in Fig 3 & 5 would suffice.

Minor comments:

-On pg 5 of the results describing the data in Supplemental Figures 1b and 2b – they allude to these data as coming from PTEN –null stroma, however, the label in the figure is ErbB2;Fsp-Cre;Pten fl/fl vs ErbB2/pten fl/fl. Please clarify.

-On pg 9 of the discussion there is a typo close to the end of the page “In breast cancer treatment, radiation is increases....”

-In the Fig 4 legend (panel a) there seems to be a mistake in the 6 hrs post-radiation in line 3 that states “DMSO vs DMSO p value..”

-On pg 7 of the results they mention that “There was a wide range of intra-sample heterogeneity, that is, even some samples with overall high stromal PTEN exhibited focal PTEN loss” The data are shown in Suppl Fig 6a. But from this panel it is not really clear that we are looking at intra-sample heterogeneity of PTEN levels. Is this a mistake?? Are they referring to the IHC panels in Fig 5b?

Reviewer #3 (Remarks to the Author):

The manuscript of Sizemore et al is part of the growing interest in the role that stroma can play in tumor progression. The aim of their study was to determine if deletion of PTEN expression in fibroblasts can modify the response of neighboring mammary epithelium to DNA damaging agent such as radiation. The study was very well designed and results obtained with their animal models are convincing.

The major finding is that deletion of PTEN expression in stroma induces a genomic instability in mammary epithelium which was associated with a higher activation of EGFR and to focal mammary lobuloalveolar hyperplasia in non-irradiated mice and in tumor development in irradiated mice. Because cell proliferation was increased, it was expected that DNA damage induced by radiation would be less efficiently repaired. Supporting the role of EGFR, mammary lobuloalveolar hyperplasia was abrogated by treating the animals with the EGFR inhibitor erlotinib, which also reduced the quantity of DNA double strand breaks.

The potential effect of PTEN deficient stroma on tumor growth could have an important impact on managing radiotherapy in cancer patients. We agree with the authors that a radiation dose of few Gy is associated with a higher relative risk of developing cancer in pediatric patients, and in premenopausal women for breast cancer. However, most women treated for breast cancer are in the postmenopausal stage and the proliferative activity of their breast epithelium is greatly diminished, thereby reducing the risk of secondary cancer. The association between radiotherapy and incidence of a secondary cancer or recurrence in postmenopausal women is still controversial. Also, radiotherapy does not always eradicate all cancer cells scattered within the breast and increasing the radiation dose is not an option as it may cause undesired long term complications to normal tissues. Consequently, the local recurrence is believed to origin from cancer cells that remain in the breast after treatment. It would be appropriate if the authors can mention in their discussion how PTEN deletion in stroma could affect the proliferation of these residual cancer cells, and then may promote a recurrence.

Minor points

Because whole body irradiation can induce systemic effects, it would have been preferable to irradiate only a single mammary gland in mouse models. In addition, a more relevant irradiation planning could have been carried out.

What is the meaning of MaSC?

This sentence needs to be clarified: "In breast cancer treatment, radiation increases cancer risk in the contralateral breast^{38, 39}, and pediatric patients treated with chest radiation for Hodgkin's lymphoma have a dramatically increased risk for developing aggressive breast cancer⁴⁰⁻⁴⁵, with an estimated 35% developing bilateral disease at a much younger age compared to the general population^{40, 43, 46}." What do authors mean by "an estimated 35% developing bilateral disease". A relative risk increased by 35%?

The text is well written, excepted for some typo errors, such as the space after %.
Axis of figures 1 and supplemental figure 1 are too small and difficult to read.

Response to Reviewers' comments:

Please find associated with this letter our revised manuscript entitled “Stromal PTEN determines mammary epithelial response to radiotherapy” by Sizemore *et al.* that we are resubmitting for publication in *Nature Communications* (NCOMMS-17-07590). We thank you and the reviewers for considering this body of work and for your critical assessment of our manuscript. While the reviewers had favorable comments, they did make several suggestions for revision. We have addressed each of these suggestions through additional experimental data, text additions, or through discussion below. Overall, we believe this revision has substantially improved the manuscript, and it is now suitable for publication in *Nature Communications*.

Reviewer #1

1. Authors have used fibroblast specific-1 promoter (Fsp1) for Cre expression - which should induce Cre expression specifically in the fibroblasts (as shown by authors in their previously published work). However, there is always a concern for slight 'leakiness'....We recommend the following experiments (or a combination of them) to help demonstrate that these effects are driven solely by loss of PTEN in the stroma:

a) It would help if the authors FACS sort the epithelial cell population, and demonstrate via western blot that PTEN levels are not affected in the epithelial cells and also carry out PCR analysis of the floxed allele in the epithelial cells to show that there is absolutely no deletion of PTEN floxed alleles in these epithelial cells.

Prior published work by our group has shown that *Fsp-cre* activity is not detectable in mammary epithelium [Trimboli *et al. Cancer Res.* 68:937-945 (2008), Fig. 1 and 2; Trimboli *et al. Nature* 461:1084-1091 (2009), Supp. Fig. 2], mammary associated macrophages (Trimboli 2009 - Supp. Fig. 2) or mammary associated endothelium (Trimboli 2009 - Supp. Fig. 2). In these previous publications, we also showed that PTEN protein expression is maintained in the mammary epithelium (Trimboli 2009 - Supp. Fig. 3) and in tumor epithelium emanating from *ErbB2;Fsp-cre;Pten^{fl/fl}* mice (Trimboli 2009 - Supp. Fig. 5). Despite this published evidence, we completely agree with the reviewer that any “leakiness” of the cre in the epithelial population could confound our results, and we have now included additional information that rigorously excludes this possibility.

Since our current work focuses on purified epithelium by FACS, we did as the reviewer requested, and took the FACS sorted mature luminal population (CD24⁺/CD29^{lo}/CD61⁻ cells as depicted in Figure 1d) and cytospun the purified population for PTEN IF staining. The IF staining is preferable to a western blot because we could then dual stain for cytokeratin 8, an established marker of the mammary luminal lineage, to assure a pure luminal population on a single cell level. As shown now in **Supplemental Figure 2a**, there is no change in total PTEN protein in the mature luminal epithelium harvested from *ErbB2;Pten^{fl/fl}* mice versus the *ErbB2;Fsp-cre;Pten^{fl/fl}* mice. This IF experiment was done on pooled epithelium from 3 mice per genotype. We have also added supporting data to show there is no change in *Pten* mRNA expression between sorted mature luminal epithelia harvested from the *ErbB2;Pten^{fl/fl}* versus *ErbB2;Fsp-cre;Pten^{fl/fl}* mice in 5 additional sorting experiments of

pooled mice. These data are now in **Supplemental Figure 2b**. Combined, these data are discussed on pages 4-5 of the results.

To further support the absence of *Fsp-cre* leakiness, we performed three additional experiments on the four independent tumors that arose from transplantation of sorted *ErbB2;Fsp-cre;Pten^{fl/fl}* epithelium represented in Figure 1b and 1c:

1. We ran genotyping PCR analysis that detects wild-type, floxed and deleted *Pten* alleles on genomic DNA isolated from the tumors. Now shown in **Supplemental Figure 1b**, the 300bp floxed allele is prominent in each of these tumors as expected as the donor mouse was *Pten^{fl/fl}*. The presence of a faint wild-type 220bp band is also not surprising given infiltration of host wild-type stroma/immune cells that were inevitably included in the total tumor genomic DNA preparation. Most importantly, there is no detection of the deleted 280bp *Pten* allele. Knowing that the deleted band could be present at a much lower abundance versus the floxed band, we also ran the PCR using the two primers that ONLY detect the deleted allele to be sure that this band is truly not detectable. These results are also shown in **Supplemental Figure 1b**.
2. We isolated total RNA from the four tumors and ran qRT-PCR for *Cre* and *Pten* mRNA expression (now **Supplemental Figure 1c**). In support of a stromal-specific *Fsp-cre*, *Cre* mRNA expression is completely undetectable and *Pten* mRNA expression is detectable in all four tumors. In both cases, we compare mRNA levels relative to isolated WT control (*Pten^{fl/fl}*) and PTEN-null (*Fsp-cre;Pten^{fl/fl}*) mouse mammary fibroblasts (MMFs).
3. Since the donor tissue expresses LacZ under the Rosa26 promoter as a readout of the Cre, we performed both PTEN IHC and LacZ staining on the four tumors visualizing the maintenance of PTEN protein and the absence of LacZ throughout the tumor tissue. Representative images shown in **Supplemental Figure 1d**.

If rare activity of the *Fsp-cre* transgene in our sorted epithelial populations occurred, we would expect the opposite results. Combined, these data confirm that tumors did not arise due to leakiness of *Fsp-cre* in the epithelial compartment.

b) Instead of injecting the PTEN positive epithelial cells (from mice which had PTEN null stroma), it will be more powerful to take the PTEN null stroma, devoid of any epithelial cells, and then inject those in the wt mice. Others have used this approach successfully to study the effect of stroma on mammary tumorigenesis. If injecting PTEN null stroma induces tumor formation in wt mice – this will go a long way in confirming the conclusions put forth by the authors.

The reviewer is correct in that co-injecting “pro-tumorigenic” stroma along with established tumor cells leads to an increase in tumorigenic properties [Orimo *et al. Cell* 121:335-348 (2005)]. In these experiments the fibroblasts are co-injected with the tumor cells in a single bolus. This methodology works to test stromal contribution to tumorigenesis for three reasons: 1) the co-injection forces the test tumor cells to interact directly with the test stromal cells, 2) the tumor cells are already transformed, and 3) the tumor cells are in great abundance. In contrast, we are unaware of any published literature where investigators injected modified stroma into a wild type animals and observed transformation of normal epithelium. It is known that fibroblasts in these co-injection studies are replaced by host fibroblasts within a few weeks. Thus, we believe it is technically impossible to assure that injected fibroblasts will remain and interact with the host epithelium in the wild type tissue for a long enough period to induce tumorigenesis in normal epithelium, especially given our hypothesis where paracrine EGF ligands need to activate neighboring epithelium. Even if injected PTEN null stroma was capable of transforming “normal epithelium” upon injection, additional studies would be warranted to prove that the injected cells induced transformation directly through EGF signaling (i.e. genetic manipulation of EGF ligands in the stroma, genetic manipulation of EGFR in the host epithelium). These experiments are far beyond the scope of the current work.

c) Another approach the authors could use is to transplant labeled (wt) epithelial cells in *ErbB2;Fsp-cre;Pten^{fl/fl}* and *ErbB2;Pten^{fl/fl}* mice; allow them to grow in the presence of PTEN-depleted stroma and then FACS sort these labeled cells and now inject these cells in wt mice (as done for Fig. 1b). If these

cells form tumors in the recipient mice – there will be no doubt that these PTEN wt cells which grew in presence of PTEN null stroma are transformed. The authors can also look for signs of genomic instability in these ‘labeled’ wt epithelial cells that have grown in presence of Pten null stroma (vis-a-vis Fig 2c).

The experiments described in point 1b and 1c are good suggestions by the reviewer if there is concern of “leakiness” of our *Fsp-cre*, but are technically challenging and remain subject to alternate interpretations even if successful. We believe that the additional data we have provided for point 1a and point 2 (see below) convincingly shows that the results are not due to rare deletion of *Pten* and subsequent selection. Given these data, we hope the reviewers agree these complex experiments are not worth the resources, in particular the number of additional animals required, and substantial time required to perform the experiments properly.

2. MMTV-ErbB2 is an established mouse model to study mammary tumorigenesis. There is a concern that in ErbB2;*Fsp-cre*;*Pten^{f/f}* mice, ErbB2 dependent tumorigenesis (irrespective of PTEN null stroma) could induce EMT which would essentially mean that the PTEN floxed epithelial cells in ErbB2;*Fsp-cre*;*Pten^{f/f}* mice undergoing early EMT- associated changes will now start expressing Cre leading to loss of PTEN from the epithelial cells.

We completely agree with the reviewer that the development of EMT in *ErbB2*;*Fsp-cre*;*Pten^{f/f}* mammary tissue could result in loss of epithelial PTEN and confound our results. In addition to the evidence we have provided to address potential *Fsp-cre* leakiness for reviewer point 1a above, we also have added discussion regarding our previous work published in *Cancer Research* in 2008 (Trimboli *et al.*, “Direct Evidence for Epithelial-Mesenchymal Transitions in Breast Cancer”) to the results section (p. 4). In this previous publication, we evaluated tumors arising in *ErbB2*;*Fsp-cre*;*Rosa^{loxP}* mice for EMT of the tumor epithelium through X-gal staining (shown in Figure 4c of that publication). For all the tumors we evaluated in this previous study (total of 78 tumors), only 4 *ErbB2*;*Fsp-cre*;*Rosa^{loxP}* tumors (5% of the total) indicated EMT (data discussed in Table 1 of that publication). This lack of EMT is in direct contrast to what was observed when *Myc* was the driving oncogene. Combined with the absence of LacZ positivity in the tumors discussed in the current manuscript (now shown in Supp. Figure 1d), we find it extremely unlikely that EMT is driving deletion of PTEN in this system.

Furthermore, regarding our current findings where we found radiation induces hyperplasia in the *Fsp-cre*;*Pten^{f/f}* mammary glands (Figure 6), we had provided representative photos of the maintenance of PTEN IHC in Figure 6C as well as the absence of epithelial LacZ staining in Supplemental Figure 8b in the hyperplastic epithelial regions. In response to this reviewer comment, we have now added an additional three photos of PTEN IHC from three additional mice in **Supplemental Figure 8a**. We did not observe any measurable loss of PTEN in the *Fsp-cre*;*Pten^{f/f}* epithelium or any significant presence of LacZ positivity in the *Fsp-cre*;*Pten^{f/f}* epithelium in any of the tissues examined.

3. Fig. 1f and Fig.2: It has been well documented now (Loonstra, Jos Jonkers et al. PNAS 2001, Marc Schmidt-Supprian & Klaus Rajewsky, Nature Immunology, 2014) that expression of Cre is DNA damage inducing independent of its activity on floxed DNA. Authors show that only the epithelial cells from ErbB2;*Fsp-cre*;*Pten^{f/f}* mice show DNA damage and not those from ErbB2;*Pten^{f/f}* mice. However, this could also be because of Cre expression in stroma of ErbB2;*Fsp-cre*;*Pten^{f/f}* mice which in turn could result in high DNA damage in stroma/fibroblasts resulting in DNA damage induced signaling and effects on the surrounding epithelial cells. This could be PTEN loss independent. An important control missing here is stroma that is expressing Cre but does not have floxed PTEN. It is a better control than the ones used by the authors (no Cre but floxed Pten alleles). It will help to rule out that the DNA damage effect - including chromosomal abnormalities are not simply because of DNA damage induced by Cre recombinase.

To address this reviewer concern, we have generated mice that are only *Fsp-cre* or wild-type. These mice were irradiated and γ -H2AX was evaluated at 6 hours post radiation similar to the experiment described in Figure 2a. Data now included in **Supplemental Figure 3d** and discussed in the results (p. 6) shows that the

presence of *Fsp-cre* alone does not significantly increase the DNA damage occurring in the mammary epithelium.

Another piece of information missing here is the status of DNA damage in the stromal cells. Is there increased DNA damage in the stroma of ErbB2;Fsp-cre;Pten^{f/f} mice as well? And what is the status of DNA damage in stromal cells which are expressing Cre but do not have the floxed PTEN?

This is an intriguing question by the reviewer, and the simple answer is yes, the loss of stromal PTEN significantly increases DNA damage in the stroma in response to radiation. These data are now included in **Supplemental Figure 3c**. Importantly, we must note that the level of DNA damage occurring in the stroma is only a fraction of what we see in the epithelium of the same tissue (*i.e.* the stromal *Pten^{f/f}* H-score mean = 4.5, versus the epithelial *Pten^{f/f}* H-score mean = 69.6). That said, this low level increase in DNA damage is likely functionally relevant. We now discuss these findings on page 6 of the results section, which are certainly not surprising based on known cell-autonomous roles of PTEN on genomic instability.

4. Fig. 1f. What are the Rad51 protein levels by western blot. Is Rad51 downregulated as suggested by the GSEA analysis?

Rad51 protein levels were evaluated by immunofluorescence in Figure 1F for multiple reasons. First, we simply do not generate enough epithelium through our FACS isolation for a western blot. Second, immunofluorescence staining allowed us to evaluate dual keratin 8 positivity which confirms the cells with the increased Rad51 foci are luminal epithelia and not stroma. And, third, we wanted to evaluate localization of Rad51 nuclear foci, which directly indicates DNA damage.

5. The authors show that the epithelial cells derived from the ErbB2;Fsp-cre;Pten^{f/f} mice but not the ErbB2;Pten^{f/f} mice give rise to tumors when transplanted into wt mice (Figure 1B) and have clear gene expression changes, suggesting that these cells have been primed for tumorigenesis by stroma depleted of Pten. Authors show that this is due to elevated EGFR signaling. To confirm this conclusion, it will help if the authors could determine *in vitro* if exogenous elevation of EGF signaling results in gene expression changes similar to those seen in Figure 1E, and will lead to similar DNA repair defects as predicted by their EGFR inhibitor assays.

In order to properly address this reviewer concern, we need to culture normal mouse mammary epithelium "*in vitro*". For the *in vitro* experiments shown in Figure 1f,g: (Day 1) bulk mammary tissue was harvested, (Day 2) epithelia was purified by gravity separation, (Day 3) epithelia was irradiated and then 6 hours later fixed for immunofluorescence staining. While these short term experiments were successful, extended culture conditions and further manipulation of the purified epithelium has been unsuccessful in our hands. We believe this technical difficulty is predominantly due to the non-transformed epithelium senescing in 2D culture conditions. Importantly, the cell viability of mouse primary mammary epithelial cells *in vitro* requires specific "epithelial" growth media supplemented with growth factors including EGF (see method section) making it technically very difficult to manipulate EGFR signaling without loss of viability. Additionally, the mammary epithelium in the stromal PTEN null mouse is exposed to a PTEN null stromal milieu since before birth. It is impossible to mimic this biology in an *in vitro* established cell line model system.

6. Given that the authors' main observation (epithelial cell transplantation expt; Fig 1b) is based on experiments carried out with ErbB2;Fsp-cre;Pten^{f/f} and ErbB2;Pten^{f/f} mice, it is not clear why the authors show half of the genomic instability data in non-ErbB2 mice (Fsp-cre;Pten^{f/f} and Pten^{f/f} mice) – especially when studying the upregulation of EGF ligands. Is this upregulation happening in mammary epithelial cells of ErbB2;Fsp-cre;Pten^{f/f} mice compared to ErbB2;Pten^{f/f} mice. It is important to know that given that the tumor studies are all done in ErbB2 mouse model.

All of the genomic instability experiments (pericentrin/ α -tubulin IF and karyotyping) were carried out in ErbB2 mice with and without stromal PTEN (Figure 2b-d; with and without Erlotinib in Figure 4c). Strikingly, the γ -H2AX DNA damage analyses look similar with or without the presence of the ErbB2 transgene (Figure 2a), and our evaluation of lobuloalveolar hyperplasia was performed in the absence of the ErbB2 transgene (Figure 6).

These are critical observations demonstrating the non-cell autonomous effect of stromal PTEN loss on genomic stability in wild-type epithelium, which may have implications in human biology given our data showing loss of PTEN expression in breast reduction samples.

Reviewer #2

1- in Fig 1 they look at effects of stromal loss of Pten on tumor formation and in the DNA repair response. Panels a and d show FACS analyses on cells from the mammary epithelium, which was analyzed then purified by CD24/CD29 positivity before being transferred into WT recipients. They mention that there are no differences in the populations. However, in a recent paper from the group (Size more et al Oncogene 2017), they show in Fig 2 that there is an expansion of the basal/myoepithelial population in the Pten null stroma. Please clarify. Could this be due to a difference in the time when the cells were harvested from the mammary glands?

There is NO difference in the CD24+/CD29+ bulk epithelial population, which is the upper right quadrant of the FACS plots shown in Figure 1a and Supp. Figure 1a. We absolutely do see an expansion in the CD24+/CD29hi “basal/myoepithelial” MaSC-enriched sub-population in the mammary glands of mice with PTEN-null stroma. This was the data published in Oncogene, 2017. To help clarify, we have added a sentence to the results sentence on p. 4.

2- in Fig 3 they show that in mammary glands that are Pten null not only EGFR is P, but also ErbB2 (panels c & d). This is not unexpected considering that EGF family ligands activate EGFR homo- and ErbB2-containing EGFR-heterodimers. In addition the NRGs also activate ErbB2 in ErbB receptor heterodimers. In Fig 4 they use the ErbB2/Pten null model for some experiments and show that chromosome breaks are lower in the erlotinib treated mice (Panel c). In panel c they show that P-EGFR levels are also lower. The data in panel b seem to be from Pten null mammary epithelium – or is this also the ErbB2/Pten null model? What about P-ErbB2 levels? Are they also affected by the EGFR inhibitor erlotinib?

As stated in the legend, the staining in Figure 4b was generated using control *Pten^{fl/fl}* mammary tissue. These mice were the experimental controls from Figure 4a, not Figure 4c. At the reviewer’s request, we performed phospho-ErbB2 immunofluorescence on these same samples. The resulting data is now shown in **Supplemental Figure 5b** and shows there is no difference in phospho-ErbB2 (Y1221/22) with erlotinib pre-treatment. We agree with the reviewer that the upregulation of phospho-ErbB2 that is depicted in Figure 3c could suggest possible heterodimer formation of EGFR/ErbB2, and that inhibition of EGFR could then possibly diminish activation of ErbB2. We have now addressed EGFR/ErbB2 heterodimerization in the results (p.7-8) and the discussion (p.11).

3- I think it would also be important to look at the impact of an ErbB2 inhibitor, in at least some of the experiments. This is particularly relevant since in the human breast cancer studies shown in Fig 5, they concentrate on patients with elevated HER-2/ErbB2. These patients will receive an inhibitor to ErbB2 and not to the EGFR. For experiments with their mouse models the ErbB2 inhibitor lapatinib would be perfect.

We had technical difficulties in achieving significant reduction in ErbB2 activation in the mammary gland at lapatinib doses (delivered by oral gavage) that were not toxic to the mice. Given these technical difficulties, we chose to use a HER2-specific inhibitor (CP-724,714) which directly addresses whether HER2 signaling is involved in the DNA damage defect we observe in the PTEN null epithelium. Treating mice with the CP-724,714 significantly inhibits ErbB2 activity, and very interestingly, abrogated the increased γ -H2AX staining observed in the mammary epithelium of mice with stromal PTEN deletion. These data are now shown in **Supplemental Figure 6** and are discussed on page 8 of the results and page 11 of the discussion.

4- Fig 5. It is interesting that CAFs and normal fibroblasts taken from an area >10cm from the tumor do show an upregulation of EGF-family ligands in response to PTEN KD (panels e & f). The particular ligands are different though. What about NRG4 that was highly upregulated in Fig 3a – was it tested?

Expression data for *NRG1*, *NRG2*, *NRG3* and *NRG4* is now included in **Supplemental Figure 7c,d** and discussed on page 9 in the results. *NRG4* was downregulated in the CAFs and not detectable in the normal fibroblasts.

Can they say anything about the mechanism downstream of PTEN loss?? What happens to the activity of MAPK and PI3K in these cells?

We have already shown that loss of PTEN hyper-activates AKT, JNK and ERK1/2 in isolated mammary fibroblasts [Trimboli *et al. Nature* 461:1084-1091 (2009), Fig. 2f] and in the mammary stroma (Trimboli 2009 - Fig. 2g). In addition, activation of JNK was shown to be critical for increased expression of Ets2, a critical factor activated by PTEN loss [Bronisz *et al. Nature Cell Biology* 14(2):159-167 (2012)].

Do the ligands share a TF binding site in their promoter that might be responsible for the transcriptional increase?

Evaluation of publicly available CHIP-seq data shows that many of these genes have binding sites for Ets-factors; however, verifying this will take several more months of experiments, and is beyond the scope of the present work.

The same question can be asked in the mouse models. In the absence of PTEN is there an increase in stromal activity of these pathways?

Answered above.

Finally, out of curiosity – the data shown in Fig 5e in which PTEN was KDd in CAFs from a breast cancer patient – did this patient’s stroma have high or low PTEN levels? Can they link overall PTEN levels this with any particular increase/or not in EGF-family ligands?

We do not have any data to suggest that levels of endogenous PTEN correlate with EGF ligands in human mammary fibroblasts.

5- While it is very likely that downregulation of P- EGFR in the mammary glands results from the kinase blocking the receptor activity, one cannot rule out the possibility that ligand expression is also decreased in response to erlotinib. In addition, the authors have only looked at ligand RNA levels and have not measured actual protein levels. Protein levels of at least some of the ligands should be examined in the CM of the fibroblasts.

We confirmed up-regulation of TGF- α by western (now **Figure 3b**). Epiregulin and amphiregulin protein up-regulation was confirmed by evaluating conditioned media isolated from wild-type (*Pten^{fl/fl}*) and PTEN-null (*Fsp-cre;Pten^{fl/fl}*) MMFs by ELISA. These data are now shown in **Supplemental Figure 4a** and discussed on page 7 of the results.

In addition the response of the fibroblasts to erlotinib treatment could be directly checked in the cultured MMFs used in Fig 3 and/or the CAFs and the fibroblasts used in Fig 5. For this latter experiment RNA levels of the ligands that they show in Fig 3 & 5 would suffice.

PTEN-null MMFs (*Fsp-cre;Pten^{fl/fl}*) were treated with and without erlotinib and harvested 24 hours later to evaluate EGF ligand mRNA expression. These data are now included in **Supplemental Figure 5a** and discussed on page 7 of the results. As the reviewer states, it is important to know if EGF ligand expression is reduced as a response to erlotinib treatment exaggerating the observed phenotype. Interestingly, the opposite occurred where the majority are increased as a response to erlotinib treatment. We predict these findings likely represent a positive feedback loop by the signaling pathway.

Minor comments:

-On pg 5 of the results describing the data in Supplemental Figures 1b and 2b – they allude to these data as coming from PTEN –null stroma, however, the label in the figure is ErbB2;Fsp-Cre;Pten fl/fl vs ErbB2/pten fl/fl. Please clarify.

The data is from *ErbB2;Fsp-Cre;Pten^{fl/fl}* vs *ErbB2;Pten^{fl/fl}*, we were speaking in a more general context that the stroma was “PTEN-null”.

-On pg 9 of the discussion there is a typo close to the end of the page “In breast cancer treatment, radiation is increases....”

Corrected.

-In the Fig 4 legend (panel a) there seems to be a mistake in the 6 hrs post-radiation in line 3 that states “DMSO vs DMSO p value..”

The comparison is between DMSO (*Pten^{fl/fl}*) vs DMSO (*Fsp-cre;Pten^{fl/fl}*). We agree with the reviewer that the shortened “DMSO vs DMSO” is confusing and have modified the text to say “DMSO (*Pten^{fl/fl}*) vs DMSO (*Fsp-cre;Pten^{fl/fl}*)”.

-On pg 7 of the results they mention that “There was a wide range of intra-sample heterogeneity, that is, even some samples with overall high stromal PTEN exhibited focal PTEN loss” The data are shown in Suppl Fig 6a. But from this panel it is not really clear that we are looking at intra-sample heterogeneity of PTEN levels. Is this a mistake?? Are they referring to the IHC panels in Fig 5b?

This was not a mistake, and we thank the reviewer for bringing to our attention the lack of clarity explaining these results. The graph in now Supplemental Fig 7a shows the minimum H-score value (in red), the maximum H-score value (in gray) and the median H-score value (in black) for each patient sample (x-axis shows all the samples 1-99). We have the data shown as line graphs because it depicts the lows and highs nicely without complicating the graph with three bars per patient (297 bars total). We have revised the legend text and have added detail regarding the figure on pages 8-9 to help clarify the results.

Reviewer #3

...local recurrence is believed to origin from cancer cells that remain in the breast after treatment. It would be appropriate if the authors can mention in their discussion how PTEN deletion in stroma could affect the proliferation of these residual cancer cells, and then may promote a recurrence.

We thank the reviewer for these thoughts and have added text to the discussion section regarding how PTEN deletion can alter neighboring residual cancer cells post treatment on page 11.

Minor points

Because whole body irradiation can induce systemic effects, it would have been preferable to irradiate only a single mammary gland in mouse models. In addition, a more relevant irradiation planning could have been carried out.

We agree with the reviewer that whole body irradiation causes systemic effects and it is absolutely more preferable to irradiate ONLY the mammary gland. Using stereotactic radiation in small animals is doable, but extremely expensive. We believe our *in vitro* data as shown in Figure 1f,g establishes that the DNA damage defect we see is not secondary to systemic effects as these cells were treated with radiation *ex vivo*.

What is the meaning of MaSC?

We apologize for not including the full description in the text. MaSC stands for mammary stem cell. This has been changed in the text on page 4.

This sentence needs to be clarified: “In breast cancer treatment, radiation is increases cancer risk in the contralateral breast^{38, 39}, and pediatric patients treated with chest radiation for Hodgkin’s lymphoma have a dramatically increased risk for developing aggressive breast cancer⁴⁰⁻⁴⁵, with an estimated 35% developing bilateral disease at a much younger age compared to the general population^{40, 43, 46}.” What do authors mean by “an estimated 35% developing bilateral disease”. A relative risk increased by 35%?

The sentence has been changed to (changes in red):

In breast cancer treatment, radiation increases cancer risk in the contralateral breast, and pediatric **female** patients treated with chest radiation for Hodgkin’s lymphoma have a dramatically increased risk for developing aggressive breast cancer, with an estimated 35% developing bilateral **breast cancer** at a much younger age compared to the general population.

The text is well written, excepted for some typo errors, such as the space after %

Corrected.

Axis of figures 1 and supplemental figure 1 are too small and difficult to read.

The size of these axes have been increased.

REVIEWERS' COMMENTS:

Reviewer #1 (Remarks to the Author):

The authors have addressed majority of the concerns - especially to do with leakiness of Cre, and the DNA damage caused by Cre expression alone.

Reviewer #2 (Remarks to the Author):

The authors did an excellent job of answering my questions and comments, as well as providing new data. The revised paper is stronger and I have no further comments.

Reviewer #3 (Remarks to the Author):

All the suggested modifications were done, and appropriate justifications were provided.